# Proximal-IMH: Proximal Posterior Proposals for Independent Metropolis–Hastings with Approximate Operators

**Youguang Chen** [1]  **George Biros** [1]

## Abstract

We are considering the problem of sampling from a posterior distribution related to Bayesian inverse problems arising in science, engineering, and imaging. Our method belongs to the family of independence Metropolis–Hastings (IMH) sampling algorithms. These are quite common in Bayesian inference. Relying on the existence of an approximate posterior distribution that is cheaper to sample from but can have significant bias, we introduce Proximal-IMH, a scheme that removes this bias: it corrects samples from the approximate posterior solving an auxiliary optimization problem, yielding a local adjustment that trades off adherence to the exact model against stability around the approximate reference point. For idealized settings, we prove that the proximal correction tightens the match between approximate and exact posteriors, and thereby improves acceptance rates and mixing. The new method works with both linear and nonlinear input-output operators and is especially suitable for inverse problems where exact posterior sampling is too expensive. We perform several numerical experiments that include multimodal and data-driven priors and nonlinear input-output operators. The results show that Proximal-IMH reliably outperforms existing IMH variants.

## 1. Introduction

We consider linear and nonlinear inverse problems of the form

$$y = \mathbf{O}\mathbf{F}(x) + e = \mathbf{A}(x) + e, \tag{1}$$

where $x \in \mathbb{R}^{d_x}$ denotes the unknown parameter vector with prior $p(x)$, $y \in \mathbb{R}^{d_y}$ is the noise-corrupted observation, and $e$ is observational noise with known distribution $q(e)$. Here, $\mathbf{O}$ is a linear *observation operator*, $\mathbf{F}(x)$ is a (possibly nonlinear) *forward operator*, and $\mathbf{A} = \mathbf{O}\mathbf{F}$ is the associated *input–output operator*. The operator $\mathbf{F}$ typically corresponds to the solution map of a differential or integral equation driven by the parameter $x$, and $u = \mathbf{F}(x)$ is a latent variable representing this solution. As an example from acoustics, $x$ may describe an unknown scatterer, $u$ is the resulting acoustic field—defined everywhere; and $y$ consists of measurements collected at sensor locations. The Bayesian posterior distribution is given by $\pi(x \mid y) \propto q(y - \mathbf{O}\mathbf{F}(x)) \, p(x)$. Our goal is to design fast and scalable algorithms for efficiently generating samples from $\pi(x \mid y)$.

Inverse problems of the form Equation (1) arise in a wide range of applications, including tomography (Kaipio et al., 2000; Saratoon et al., 2013; Arridge & Schotland, 2009), medical imaging (Liang et al., 2020; Epstein, 2007), geophysics (Tarantola, 2005), and many others (Kaipio & Somersalo, 2005; Vogel, 2002; Ghattas & Willcox, 2021). In many such problems, evaluating the forward operator $\mathbf{F}$ is computationally expensive, but a cheaper approximation $\widetilde{\mathbf{F}}$ is available such that $\widetilde{\mathbf{F}}(x) \approx \mathbf{F}(x)$. Defining $\widetilde{\mathbf{A}} = \mathbf{O}\widetilde{\mathbf{F}}$, this naturally leads to the *approximate posterior* $\pi_a(x \mid y) \propto q(y - \widetilde{\mathbf{A}}(x)) \, p(x)$. Examples of $\widetilde{\mathbf{F}}$ include preconditioners, truncated iterative solvers, coarse-grid discretizations, and learned surrogate or neural operators (Cohen et al., 2014; Van der Vorst, 2003; Henson & Yang, 2002; Herrmann et al., 2024). Thus, the approximate-operator setting considered here also covers modern surrogate-based approaches in which a cheaper learned model is used in place of the expensive forward operator.

A broad class of methods has been developed for sampling from $\pi(x \mid y)$, including Markov chain Monte Carlo (MCMC) methods such as Langevin methods (Girolami & Calderhead, 2011), NUTS (Hoffman et al., 2014), and Metropolis adjusted Langevin algorithm (MALA) (Roberts & Tweedie, 1996), generative models (Song et al., 2021), normalizing flows (Papamakarios et al., 2021), low-rank approximations (Spantini et al., 2015), and many others (Biegler et al., 2010).

Several methods seek to exploit the availability of an ap-

---

[1]Oden Institute for Computational Engineering and Sciences, The University of Texas at Austin, Austin, Texas, USA. Correspondence to: Youguang Chen <youguang@utexas.edu>.

*Proceedings of the 43rd International Conference on Machine Learning*, Seoul, South Korea. PMLR 306, 2026. Copyright 2026 by the author(s).

proximate forward operator $\widetilde{\mathbf{F}}$ to reduce computational cost. Among them, two approaches are particularly relevant to this work: Latent-IMH and multifidelity sampling. Multifidelity sampling methods, including delay-acceptance and two-stage algorithms (Peherstorfer et al., 2018), use $\widetilde{\mathbf{F}}$ to screen or precondition proposals from a local MCMC kernel. While this reduces the number of expensive forward evaluations, the resulting schemes remain inherently local, with mixing governed by the underlying proposal. Latent-IMH (Chen & Biros, 2026) constructs a global independence proposal that preserves the exact likelihood while modifying the prior via an approximate operator, enabling efficient global moves. However, it requires transforming the forward operator into a square, invertible form and constructing an approximate distribution in a high-dimensional latent space, which can lead to ill-conditioning. We review Latent-IMH in Section 2 and include it as a baseline in our numerical experiments.

In this paper, we introduce an IMH sampler that exploits $\widetilde{\mathbf{F}}$ and addresses key limitations of `Latent-IMH`. Our contributions are summarized below. ❶ We introduce the `Proximal-IMH` sampling method for inverse problems of the form Equation (1) that exploits a computationally cheap approximate operator $\widetilde{\mathbf{F}}$, and applies to both linear (Section 3.1) and nonlinear (Section 3.4) forward models. ❷ For linear forward models, we provide a theoretical analysis showing that `Proximal-IMH` achieves smaller expected KL divergence (Section 3.2) and faster mixing times (Section 3.3) than existing IMH proposals under suitable regularity assumptions. ❸ We demonstrate the effectiveness of `Proximal-IMH` on a range of linear and nonlinear inverse problems, including bimodal targets, imaging problems with learned priors, and nonlinear acoustic scattering (Section 4). Across all tests, `Proximal-IMH` consistently outperforms other IMH schemes in terms of acceptance rates, convergence speed, and sampling accuracy.

## 2. Background

We begin by defining the exact posterior and two approximate posteriors.

$$\text{Exact:} \quad \pi(x \mid y) \propto q(y - \mathbf{A}x)\, p(x), \quad (2a)$$

$$\text{Approx:} \quad \pi_a(x \mid y) \propto q(y - \widetilde{\mathbf{A}}x)\, p(x), \quad (2b)$$

$$\text{Latent:} \quad \pi_l(x \mid y) \propto q(y - \mathbf{A}x)\, p(\widetilde{\mathbf{F}}^{-1}\mathbf{F}x), \quad (2c)$$

where the formulation for `Latent posterior` assumes that $\mathbf{F}$ and $\widetilde{\mathbf{F}}$ are linear and invertible. The approximate distributions $\pi_a$ and $\pi_l$ can be used as proposal densities $g(x \mid y)$ in the independence Metropolis–Hastings (IMH) algorithm (Algorithm 1).

A key advantage of IMH over local MCMC methods is that proposals are independent of the current state. Consequently,

---

**Algorithm 1** Independence Metropolis–Hastings (IMH)

**In:** $y$, number of MH steps $k$, proposal $g(x \mid y)$
**Out:** Samples $\{x_i\}_{i=1}^{k+1}$ from $\pi(x \mid y)$
1: $x_1 \sim g(x \mid y)$
2: **for** $t = 1$ to $k$ **do**
3:      $x' \sim g(x \mid y)$
4:      $a(x', x_t) = \min\left\{1, \frac{\pi(x'|y)}{\pi(x_t|y)} \frac{g(x_t|y)}{g(x'|y)}\right\}$
5:      Accept $x'$ with probability $a(x', x_t)$
6:      **otherwise** set $x_{t+1} \leftarrow x_t$
7: **end for**

---

the computationally expensive computations involving the forward operator $\mathbf{F}$—including those required for proposal construction and the corresponding terms in the acceptance ratio—can be precomputed in advance and evaluated in an embarrassingly parallel fashion. In practice, this can lead to substantial wall-clock savings for large-scale inverse problems. The efficiency of IMH is therefore governed by the acceptance ratio: for a fixed accuracy target, lower acceptance rates require more Metropolis–Hastings steps and hence more evaluations of the exact forward operator, increasing the overall computational cost. Using `Approx posterior` as the proposal is computationally attractive, but for Gaussian noise the log-acceptance ratio scales as $1/\sigma^2$, causing small operator errors in $\widetilde{\mathbf{A}}$ to be strongly amplified and leading to poor acceptance rates in practice.

The `Latent-IMH` method (Chen & Biros, 2026) addresses this issue by preserving the exact likelihood $q(y - \mathbf{A}x)$ in the proposal distribution. This is achieved by sampling in the latent variable $u = \widetilde{\mathbf{F}}x$. Specifically, samples are first drawn from an approximate latent prior $\widetilde{p}(u)$ constructed offline, then conditioned on $y$ using only the inexpensive observation operator $\mathbf{O}$, and finally mapped back via $x = \mathbf{F}^{-1}u$. This procedure is equivalent to using Equation (2c) as the IMH proposal. While effective, `Latent-IMH` requires squaring $\mathbf{F}$ to enforce invertibility, which can introduce ill-conditioning, and relies on constructing the approximate distribution $\widetilde{p}(u)$.

## 3. Proximal Independent Metropolis–Hastings

The motivation for `Proximal-IMH` is to provide an alternative to `Latent-IMH` that avoids ill-conditioning induced by squaring $\mathbf{F}$ and bypasses sampling in the latent variable $u$, which can be high-dimensional, nonlinear, or degenerate in practice. `Proximal-IMH` instead proceeds in two steps:

$$\begin{cases} \widetilde{x} \sim \pi_a(x \mid y), & (3a) \\ x = \underset{x}{\arg\min}\, \|\mathbf{A}(x) - \widetilde{\mathbf{A}}(\widetilde{x})\|^2 + \beta\|x - \widetilde{x}\|^2. & (3b) \end{cases}$$

## 3.1. Linear Input-Output Operator $\mathbf{A}$

When the forward operator $\mathbf{A}$ is linear and full rank, the optimization problem Equation (3b) admits a closed-form solution:

$$x = \mathbf{K}\widetilde{x}, \qquad \mathbf{K} = (\mathbf{A}^\top \mathbf{A} + \beta \mathbf{I})^{-1}(\mathbf{A}^\top \widetilde{\mathbf{A}} + \beta \mathbf{I}). \quad (4)$$

We assume throughout that $\mathbf{K}$ is invertible. Since $\beta > 0$, $\mathbf{A}^\top \mathbf{A} + \beta \mathbf{I}$ is symmetric positive definite, and therefore $\mathbf{K}$ is invertible if and only if $\mathbf{A}^\top \widetilde{\mathbf{A}} + \beta \mathbf{I}$ is invertible, or equivalently $-\beta \notin \sigma(\mathbf{A}^\top \widetilde{\mathbf{A}})$. This is a generic condition: for fixed $\mathbf{A}$ and $\widetilde{\mathbf{A}}$, singularity can occur only for exceptional values of $\beta$, namely $\beta = -\lambda$ for a negative real eigenvalue $\lambda$ of $\mathbf{A}^\top \widetilde{\mathbf{A}}$. Hence the condition is nonrestrictive in practice and can be verified numerically.

**Computational cost.** We do not form or invert $\mathbf{K}$ explicitly. Applying the proximal correction amounts to solving the symmetric positive definite system

$$(\mathbf{A}^\top \mathbf{A} + \beta \mathbf{I})x = (\mathbf{A}^\top \widetilde{\mathbf{A}} + \beta \mathbf{I})\widetilde{x},$$

which can be solved using CG or PCG. Each CG iteration requires one application of $\mathbf{A}$ and one application of $\mathbf{A}^\top$. In PDE-based settings, this system can be preconditioned using the cheaper approximate operator, for example with $\widetilde{\mathbf{A}}^\top \widetilde{\mathbf{A}} + \beta \mathbf{I}$. Moreover, because IMH proposals are independent, these correction solves can be performed in parallel.

The resulting proposal distribution is the push-forward $\pi_p(\cdot \mid y) = \mathbf{K}_\# \pi_a(\cdot \mid y)$ and admits the density

$$\pi_p(x \mid y) \propto q\Big(y - \widetilde{\mathbf{A}}\mathbf{K}^{-1}x\Big)\, p\big(\mathbf{K}^{-1}x\big). \quad (5)$$

Accordingly, the Metropolis–Hastings acceptance ratio takes the form

$$\min\left\{1, \ \frac{q(y - \mathbf{A}x')}{q(y - \widetilde{\mathbf{A}}\widetilde{x}')}\frac{q(y - \widetilde{\mathbf{A}}\widetilde{x}_t)}{q(y - \mathbf{A}x_t)}\frac{p(x')}{p(\widetilde{x}')}\frac{p(\widetilde{x}_t)}{p(x_t)}\right\}, \quad (6)$$

where $\widetilde{x}_t, \widetilde{x}' \sim \pi_a(x \mid y)$, $x_t = \mathbf{K}\widetilde{x}_t$ is the current state, and $x' = \mathbf{K}\widetilde{x}'$ is the proposed state.

The operator $\mathbf{K}$ can be equivalently expressed as

$$\mathbf{K} = \mathbf{I} + \mathbf{A}^\dagger(\widetilde{\mathbf{A}} - \mathbf{A}), \quad (7)$$

where $\mathbf{A}^\dagger = (\mathbf{A}^\top \mathbf{A} + \beta \mathbf{I})^{-1}\mathbf{A}^\top$ is a regularized pseudoinverse of $\mathbf{A}$. Under this form of $\mathbf{K}$, the proposal mechanism of Proximal-IMH (Equations (3a) and (3b)) can be interpreted as a correction of an approximate posterior sample $\widetilde{x} \sim \pi_a(x \mid y)$ via

$$x \leftarrow \widetilde{x} + \mathbf{A}^\dagger(\widetilde{\mathbf{A}}\widetilde{x} - \mathbf{A}\widetilde{x}). \quad (8)$$

Here, the term $\widetilde{\mathbf{A}}\widetilde{x} - \mathbf{A}\widetilde{x}$ represents a correction of the observation residual induced by the approximate forward model, while the operator $\mathbf{A}^\dagger$ maps this correction back into the parameter (i.e., $x$-) space.

**Hyperparameter $\beta$.** The regularization parameter $\beta$ controls the trade-off between enforcing consistency with the exact forward model and stabilizing the correction around the approximate sample $\widetilde{x}$. To balance these effects relative to the noise level, we choose $\beta = \Theta(\sigma^2)$, where $\sigma^2$ is the variance of the observational noise. This scaling prevents overfitting to noise while ensuring that the correction remains effective in mitigating operator mismatch. In particular, $\beta = \Theta(\sigma^2)$ is required in the KL-divergence analysis (Section 3.2) and for the mixing-time bounds of Proximal-IMH (Theorem 3.3). We further investigate the effect of $\beta$ in the numerical experiments.

### 3.2. Comparison of Proposal Distributions

We compare the proposal distributions used in Approx-IMH, Latent-IMH, and Proximal-IMH against Exact posterior using the expected Kullback–Leibler (KL) divergence, where the expectation is taken with respect to the observation $y$. For Approx posterior, we define the expected KL divergence $\mathbb{D}_a$ as

$$\mathbb{D}_a := 2\,\mathbb{E}_y[\mathrm{KL}(\pi_a(x \mid y) \,\|\, \pi(x \mid y))]. \quad (9)$$

The quantities $\mathbb{D}_l$ and $\mathbb{D}_p$ are defined analogously for Latent posterior and Proximal posterior, respectively.

**Assumption 3.1.** The prior is $p(x) = \mathcal{N}(\mathbf{0}, \mathbf{I})$ and the observational noise is $q(e) = \mathcal{N}(\mathbf{0}, \sigma^2 \mathbf{I})$.

Under Assumption 3.1, and assuming that the forward operators $\mathbf{A}$ and $\widetilde{\mathbf{A}}$ are full-rank, all posterior distributions appearing in our analysis are multivariate Gaussian (see Table 2).

**Diagonal case.** We first consider a simplified setting in which both $\mathbf{F}$ and $\widetilde{\mathbf{F}}$ are diagonal, with an identity observation operator $\mathbf{O} = [\mathbf{I}\ \mathbf{0}]$. Closed-form expressions for the corresponding expected KL divergences are given below.

**Theorem 3.2** (Adapted from Proposition 3.3 in (Chen & Biros, 2026)). *Under Assumption 3.1, let $\mathbf{F}_{ii} = s_i$ and $\widetilde{\mathbf{F}}_{ii} = \alpha_i s_i$ for $i \in [d]$. Define $\rho_i := \alpha_i^2 s_i^2 + \sigma^2 / s_i^2 + \sigma^2$ and $\zeta_i := 1/(\alpha_i^2 s_i^2 + \sigma^2)^2$. Setting $\beta = \sigma^2$ in the Proximal proposal, the expected KL divergences are given by*

$$\mathbb{D}_a = -d_y + \sum_{i=1}^{d_y}\frac{1}{\rho_i} + \log\rho_i + \zeta_i(\alpha_i - 1)^2(\alpha_i s_i^2 - \sigma^2)^2\frac{s_i^2}{\sigma^2},$$

$$\mathbb{D}_l = -d + \sum_{i=1}^{d_y}\left(\frac{\alpha_i^2}{\rho_i} + \log\frac{\rho_i}{\alpha_i^2}\right) + \sum_{i=d_y+1}^{d}\left(\alpha_i^2 + \log\frac{1}{\alpha_i^2}\right)$$

$$+ \sum_{i=1}^{d_y}\zeta_i(\alpha_i^2 - 1)^2 s_i^2 \sigma^2.$$

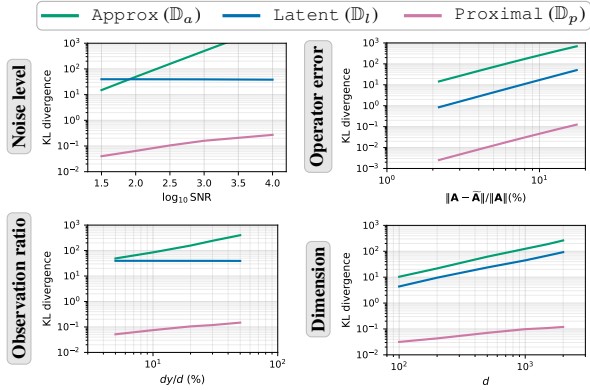

*Figure 1. Sensitivity of the expected KL divergence of the* `Approx`*,* `Latent`*, and* `Proximal` *proposals (relative to the* `Exact` `posterior`*) for different noise levels, operator errors, observation ratios, and dimensions. Details of the experimental setup are given in Table 3.*

$$\mathbb{D}_p = -d_y + \sum_{i=1}^{d_y}\left(\rho_i + \log\frac{1}{\rho_i}\right) + \zeta_i(\alpha_i - 1)^2 s_i^2 \sigma^2. \tag{10}$$

Let $\epsilon := |\alpha_i - 1|$ denote the perturbation magnitude. When $\epsilon$ is small, the leading-order behavior is given by

$$\mathbb{D}_a \sim \epsilon^2 d_y + \epsilon^2 \sum_{i\in[d_y]}\frac{s_i^2}{\sigma^2}, \quad \mathbb{D}_l \sim \epsilon^2 d + \epsilon^2 \sum_{i\in[d_y]}\frac{\sigma^2}{s_i^2},$$

$$\mathbb{D}_p \sim \epsilon^2 d_y + \epsilon^2 \sum_{i\in[d_y]}\frac{\sigma^2}{s_i^2}. \tag{11}$$

In most applications of interest, $\sum_i s_i^2/\sigma^2 \gg \sum_i \sigma^2/s_i^2$, which implies that $\mathbb{D}_a$ typically dominates both $\mathbb{D}_l$ and $\mathbb{D}_p$. Moreover, while $\mathbb{D}_l$ and $\mathbb{D}_p$ exhibit a comparable mean mismatch, the covariance mismatch of $\mathbb{D}_p$ is generally smaller, especially when $d \gg d_y$.

**General case.** Beyond the diagonal case, we study the sensitivity of $\mathbb{D}_a$, $\mathbb{D}_l$, and $\mathbb{D}_p$ in a more general setting. We consider $\mathbf{F} = \mathbf{V}\mathbf{S}\mathbf{V}^\top$ and its approximation as $\widetilde{\mathbf{F}} = \mathbf{V}\widetilde{\mathbf{S}}\mathbf{V}^\top$, where $\mathbf{V}$ is a unitary matrix obtained from the singular value decomposition of a random matrix. The diagonal matrix $\mathbf{S}$ has entries $\{1/i^2\}_{i\in[d]}$, while $\widetilde{\mathbf{S}}$ has entries $\{\alpha_i/i^2\}_{i\in[d]}$. The observation operator $\mathbf{O}$ is taken to be a full-rank random matrix. We evaluate the expected KL divergence by varying four factors: the noise level, the magnitude of the operator perturbation, the observation ratio ($d_y/d$), and the problem dimension. The experimental design for these sensitivity studies is summarized in Table 3, and the corresponding results are reported in Figure 1. In all regimes tested, $\mathbb{D}_p$ consistently achieves substantially lower values—often by orders of magnitude—than both $\mathbb{D}_a$ and $\mathbb{D}_l$.

### 3.3. Mixing Time of IMH Schemes

We now compare the convergence behavior of different IMH methods in terms of mixing time. The mixing time of the `Approx-IMH` chain under a warm start $\pi_a(x \mid y)$ is defined as

$$\tau_{mix}^a(\epsilon) := \inf\left\{n \in \mathbb{N} : \|\pi_a P_a^n - \pi\|_{\mathrm{TV}} \leq \epsilon\right\}, \tag{12}$$

where $P_a$ denotes the IMH transition kernel with the proposal distribution $\pi_a(x \mid y)$, and $\|\cdot\|_{\mathrm{TV}}$ denotes the total variation distance. Analogously, we define the mixing times for `Latent-IMH` and `Proximal-IMH` as $\tau_{mix}^l(\epsilon)$ and $\tau_{mix}^p(\epsilon)$, respectively. The theorem below establishes explicit mixing-time bounds for the three IMH schemes under the assumption of a linear forward model and a log-concave prior.

**Theorem 3.3** (Mixing time for three IMH schemes.)**.** *Assume that the prior density $p(x)$ is log-concave and that there exist constants $L, m > 0$ such that $-\log p(x)$ is $L$-smooth and $m$-strongly convex on $\mathbb{R}^d$. Assume further that the noise density $q(e)$ is Gaussian with variance $\sigma^2$, and that the forward operator $\mathbf{F}$ and its approximation $\widetilde{\mathbf{F}}$ are linear and invertible (as required by the* `Latent-IMH` *proposal). In* `Proximal-IMH`*, set the regularization parameter $\beta = \Theta(\sigma^2)$. Let $\Delta\mathbf{A} := \widetilde{\mathbf{A}} - \mathbf{A}$. Then the mixing times of* `Approx-IMH`*,* `Latent-IMH`*, and* `Proximal-IMH` *satisfy*

$$\tau_{mix}^a(\epsilon) \sim \frac{d}{m^2}\frac{\|\mathbf{A}^\top\Delta\mathbf{A}\|^2}{\sigma^4}\log(1/\epsilon), \tag{13}$$

$$\tau_{mix}^l(\epsilon) \sim \frac{dL^2}{m^2}\left\|\mathbf{I} - \widetilde{\mathbf{F}}^{-1}\mathbf{F}\right\|^2\log(1/\epsilon), \tag{14}$$

$$\tau_{mix}^p(\epsilon) \sim \frac{dL^2}{m^2}\left\|\mathbf{I} - \mathbf{K}^{-1}\right\|^2\log(1/\epsilon). \tag{15}$$

*Proof sketch.* The result follows from the general mixing-time bound in Theorem A.3, which reduces the analysis to bounding local Lipschitz constants of the log-weight functions associated with each IMH proposal. The detailed proof is provided in Section A.2. For `Approx-IMH`, the log-weight admits a quadratic form involving the operator mismatch $\Delta\mathbf{A}$, resulting in a Lipschitz constant proportional to $\|\mathbf{A}^\top\Delta\mathbf{A}\|/\sigma^2$. For `Latent-IMH`, smoothness of the negative log-prior implies that the Lipschitz constant is controlled by $\|\mathbf{I} - \widetilde{\mathbf{F}}^{-1}\mathbf{F}\|$. For `Proximal-IMH`, the correction operator $\mathbf{K}^{-1}$ introduces additional regularization, and choosing $\beta = \Theta(\sigma^2)$ ensures dependence on $\|\mathbf{I} - \mathbf{K}^{-1}\|$. Substituting these bounds into Theorem A.3 yields the stated scaling of the mixing times. $\square$

As shown in Theorem 3.3, the mixing time of `Approx-IMH` scales unfavorably with the signal-to-noise ratio $\|\mathbf{A}\|/\sigma^2$, which is typically large in inverse problems.

Both `Latent-IMH` and `Proximal-IMH` remove this explicit dependence, but differ in how operator errors are amplified.

For `Latent-IMH`, the term $\|\mathbf{I} - \widetilde{\mathbf{F}}^{-1}\mathbf{F}\|$ depends on the full perturbation $\Delta\mathbf{F} = \widetilde{\mathbf{F}} - \mathbf{F}$ and is amplified by the conditioning of $\widetilde{\mathbf{F}}$ through $\|\widetilde{\mathbf{F}}^{-1}\|$, which can become large even for small $\|\Delta\mathbf{F}\|$. In contrast, `Proximal-IMH` depends only on the observable perturbation $\Delta\mathbf{A} = \mathbf{O}\Delta\mathbf{F}$, filtered through the regularized pseudoinverse $\mathbf{A}^{\dagger}$ in $\mathbf{K} = \mathbf{I} + \mathbf{A}^{\dagger}(\widetilde{\mathbf{A}} - \mathbf{A})$. As a result, the mixing behavior of `Proximal-IMH` is more robust to operator perturbations.

A sharper comparison can be obtained in the diagonal setting introduced in the previous section for the KL-divergence comparison. In that case, one can show that

$$\|\mathbf{I} - \mathbf{K}^{-1}\| \le \|\mathbf{I} - \widetilde{\mathbf{F}}^{-1}\mathbf{F}\|, \qquad (16)$$

with equality attained only when $\mathbf{F} = \widetilde{\mathbf{F}}$.

### 3.4. Nonlinear Operator

We now extend `Proximal-IMH` to nonlinear forward problems. Recall the optimization problem Equation (3b): given a sample $\widetilde{x} \sim \pi_a(x \mid y)$ from the approximate posterior, it is expected that the proposal $x$ will remain close to $\widetilde{x}$ while reducing the data misfit. In the nonlinear least-squares setting, Gauss–Newton methods are known to achieve local quadratic convergence when the residual is small, making them a natural choice for this correction step.

Specifically, we consider a single Gauss–Newton step applied to Equation (3b),

$$x \leftarrow GN(\widetilde{x}) := \widetilde{x} - \left(\mathbf{J}(\widetilde{x})^{\top}\mathbf{J}(\widetilde{x}) + \beta\mathbf{I}\right)^{-1} \mathbf{J}(\widetilde{x})^{\top} r(\widetilde{x}), \tag{17}$$

where $r(\widetilde{x}) = \mathbf{A}(\widetilde{x}) - \widetilde{\mathbf{A}}(\widetilde{x})$ denotes the residual and $\mathbf{J}(\widetilde{x})$ is the Jacobian of $\mathbf{A}(\cdot)$ evaluated at $\widetilde{x}$. When $\mathbf{A}$ is linear, a single Gauss–Newton step recovers the exact solution given in Equation (4).

Under this construction, the `Proximal-IMH` proposal induced by one Gauss–Newton step can be expressed as a pushforward distribution,

$$\pi_p(x \mid y) = \pi_a(\widetilde{x} \mid y)\,|\det \mathbf{J}_{GN}(\widetilde{x})|^{-1}, \qquad (18)$$

where $\widetilde{x} = GN^{-1}(x)$ and $\mathbf{J}_{GN}(\widetilde{x})$ denotes the Jacobian of the Gauss–Newton map $GN(\cdot)$ at $\widetilde{x}$. The corresponding Metropolis–Hastings acceptance ratio is

$$\min\left\{1,\ \frac{\pi(x' \mid y)}{\pi(x_t \mid y)}\frac{\pi_a(\widetilde{x}_t \mid y)}{\pi_a(\widetilde{x}' \mid y)}\frac{|\det \mathbf{J}_{GN}(\widetilde{x}')|}{|\det \mathbf{J}_{GN}(\widetilde{x}_t)|}\right\}, \qquad (19)$$

where $\widetilde{x}_t, \widetilde{x}' \sim \pi_a(x \mid y)$ and $x_t = GN(\widetilde{x}_t)$ denote the current state, $x' = GN(\widetilde{x}')$ is the proposed state.

Compared with the linear acceptance ratio in Equation (6), the only additional term is the Jacobian determinant ratio. Since the Gauss–Newton update Equation (17) represents a small, local correction around $\widetilde{x}$, we expect this determinant ratio to be close to one in practice and thus to have a negligible impact on the acceptance ratio.

To illustrate this setting concretely, we consider a nonlinear model $y = \mathbf{A}G(x) + e$, where $\mathbf{A}$ is a full-rank matrix and $G$ is a nonlinear mapping. We assume a Gaussian prior on $x$ and additive Gaussian noise $e \sim \mathcal{N}(0, \sigma^2\mathbf{I})$. This model is standard in inverse problems with deep generative model–informed priors, including generative adversarial networks (GANs), variational autoencoders (VAEs), and normalizing flows, where $x$ represents the latent variable and $G$ corresponds to the decoder or generator implemented by a neural network.

Consider a small perturbation $\Delta\mathbf{A} = \widetilde{\mathbf{A}} - \mathbf{A} = \delta\mathbf{A}$ for some $\delta \in (0, 1)$. For arbitrary points $\widetilde{x}_1$ and $\widetilde{x}_2$, we have

$$\log \frac{|\det \mathbf{J}_{GN}(\widetilde{x}_1)|}{|\det \mathbf{J}_{GN}(\widetilde{x}_2)|} = \sum_{i\in[d_x]} \log \frac{1 - \delta\lambda_{1,i}/(\lambda_{1,i} + \beta)}{1 - \delta\lambda_{2,i}/(\lambda_{2,i} + \beta)}, \tag{20}$$

where $\{\lambda_{1,i}\}$ and $\{\lambda_{2,i}\}$ denote the eigenvalues (in descending order) of $\mathbf{J}^{\top}(\widetilde{x}_1)\mathbf{J}(\widetilde{x}_1)$ and $\mathbf{J}^{\top}(\widetilde{x}_2)\mathbf{J}(\widetilde{x}_2)$, respectively. Since $\beta = \Theta(\sigma^2)$, when $\lambda_{1,i}, \lambda_{2,i} \gg \beta$ the corresponding terms are negligible. When $\lambda_{1,i}$ and $\lambda_{2,i}$ are comparable to $\beta$, the log ratio is determined by their difference but remains uniformly bounded by $\log(1 - \delta)^{-1}$.

In our nonlinear implementation, we neglect the Jacobian determinant ratio when computing the acceptance probability. As a result, the nonlinear implementation should be viewed as an approximate MH scheme and may introduce bias. Exactness can be recovered by including the determinant ratio in the acceptance probability, at additional computational cost. In large-scale settings, this cost may be reduced using randomized matrix-free trace or log-determinant estimators (Saibaba et al., 2017; Knoll & Keyes, 2004).

We use one Gauss–Newton step as a practical default. For more nonlinear problems, or when the approximate operator is less accurate, additional Gauss–Newton steps may improve proposal quality, but at increased per-proposal cost. Thus, the number of correction steps introduces a cost–accuracy tradeoff.

In practice, both choices can be guided by a short pilot run. The determinant ratio can be monitored on pilot samples to assess whether it deviates substantially from one and should be included in the acceptance probability. Similarly, the number of Gauss–Newton steps can be selected by monitoring the acceptance rate and the proximal objective in Equation (3b).

# 4. Numerical Experiments

We design numerical experiments to compare `Proximal-IMH` with existing IMH schemes, evaluate its nonlinear extension, and assess consistency with the theory. Performance is measured using acceptance rates, relative mean error, and relative componentwise second-moment error. In all experiments, the observational noise is Gaussian and the noise-to-signal ratio $\|e\|/\|y\|$ is controlled between 15% and 20%.

We additionally include the No-U-Turn Sampler (NUTS) as a baseline for sampling quality and computational efficiency. Since NUTS requires repeated evaluations of the exact forward operator, it is used primarily as a benchmark rather than a directly comparable low-cost alternative. Table 1 summarizes the qualitative properties and per-sample cost model of the IMH-based methods and NUTS. In our cost accounting, we count exact forward and adjoint solves and treat solves involving the approximate operator as lower-cost overhead, reflecting the regime where the exact forward model is the dominant computational bottleneck.

We present a linear forward model test in Section 4.1 and two nonlinear forward model tests in Sections 4.2 and 4.3.

## 4.1. Bimodal Test

We consider the following prior distribution

$$p(x) \propto \exp\left( -\frac{1}{2}\|x\|^2 - \tau(w^\top x - c)^2 (w^\top x + c)^2 \right),$$
(21)

where $x \in \mathbb{R}^{200}$, $w \in \mathbb{R}^{200}$ satisfies $\|w\| = 1$, and we set $c = 2$ and $\tau = 0.3$. This prior is bimodal along the projection direction $w^\top x$. We visualize the marginal prior distribution of $w^\top x$ in Figure 10.

We construct $\mathbf{A} = \mathbf{OF}$ with $\mathbf{F} = \mathbf{VSV}^\top$, where $\mathbf{V}$ is orthogonal, $\mathbf{S}$ is diagonal with $\mathbf{S}_{ii} = 1/i$, $d_y = 50$, and $\mathbf{O} \in \mathbb{R}^{d_y \times d_x}$ is well conditioned.

We consider three different constructions of the approximate forward operator $\widetilde{\mathbf{F}}$, with $\widetilde{\mathbf{A}} = \mathbf{O}\widetilde{\mathbf{F}}$:

- **Test I (Multiplicative spectral perturbation).** $\widetilde{\mathbf{F}} = \mathbf{V}\widetilde{\mathbf{S}}\mathbf{V}^\top$, where $\widetilde{\mathbf{S}}$ is diagonal with entries $\widetilde{\mathbf{S}}_{ii} = \alpha_i \mathbf{S}_{ii}$, $\alpha_i \sim \mathcal{U}[\alpha_-, \alpha_+]$, with $\alpha_- \in (0,1)$ and $\alpha_+ > 1$.
- **Test II (Additive low-rank perturbation).** We define $\widetilde{\mathbf{F}} = \mathbf{F} + \epsilon \mathbf{U}_1 \mathbf{U}_2^\top$, where $\mathbf{U}_1, \mathbf{U}_2 \in \mathbb{R}^{d_x \times 5}$ have i.i.d. standard normal entries.
- **Test III (Low-rank approximation).** We define $\widetilde{\mathbf{F}} = \mathbf{V}\widetilde{\mathbf{S}}\mathbf{V}^\top$, where $\widetilde{\mathbf{S}}$ is obtained by setting $\widetilde{\mathbf{S}}_{ii} = \mathbf{S}_{ii}$ if $\mathbf{S}_{ii} > \alpha$ and $\widetilde{\mathbf{S}}_{ii} = 0$ otherwise, for a threshold $\alpha > 0$.

Figure 2 compares the sampling performance of different methods in the bimodal test between Tests I–III. Across

Tests I–III, `Proximal-IMH` consistently outperforms `Approx-IMH` and `Latent-IMH`, achieving substantially higher acceptance rates, particularly as $\|\mathbf{A} - \widetilde{\mathbf{A}}\|/\|\mathbf{A}\|$ increases.

In Test II, the additive low-rank perturbation $\Delta\mathbf{F}$ significantly degrades the conditioning of $\mathbf{F}$, leading to amplified inverse-operator errors. To interpret this behavior, we report in Figure 3 the operator discrepancy terms that appear in the mixing-time bounds Equations (14) and (15). Although the theoretical analysis in Section 3.3 assumes log-concave targets and the bimodal prior used here violates this assumption, the same operator-norm quantities still govern proposal sensitivity. In particular, the filtered error $\|\mathbf{I} - \mathbf{K}^{-1}\|$ remains stable under perturbations that cause $\|\mathbf{I} - \widetilde{\mathbf{F}}^{-1}\mathbf{F}\|$ to grow rapidly, which explains the improved robustness of `Proximal-IMH` relative to `Latent-IMH`.

From the convergence plots in Figure 2, `Proximal-IMH` converges significantly faster than the other IMH variants and achieves accuracy comparable to NUTS while retaining independent proposals, which can be generated and evaluated in parallel. In these plots, the x-axis counts proposals for IMH-based methods and exact forward solves for NUTS. To account for the cost of the inner linear solves, we also report error versus the number of exact forward/adjoint solves in Figure 11. This comparison still shows a clear advantage for `Proximal-IMH` over the other IMH variants and competitive accuracy relative to NUTS. The comparison is conservative for `Proximal-IMH`, since we use unpreconditioned CG; effective preconditioning can further reduce the cost.

The right columns show histograms of the projected samples $w^\top x$ for two representative perturbation levels. `Approx-NUTS`, which samples the approximate posterior $\pi_a(x \mid y)$, exhibits biased mode weights, whereas `Proximal-IMH` accurately captures both modes in all tests.

Finally, Figure 4 presents the effect of the hyperparameter $\beta$ in `Proximal-IMH`. In all tests, higher acceptance rates correlate with lower relative mean errors, with optimal performance consistently achieved near $\beta = \sigma^2$. Performance degrades noticeably for $\beta > 2\sigma^2$, indicating increased sensitivity to over-regularization.

## 4.2. MNIST Test

We consider an inverse imaging problem with the forward model $y = \mathbf{A}x + e$, where $x$ denotes the image in pixel space, $\mathbf{A}$ is a full-rank linear operator, and $y$ is the observed signal. As a prior, we use the decoder of a variational autoencoder (VAE) trained on the MNIST dataset (LeCun et al., 1998). Specifically, let $G(\cdot)$ denote the decoder mapping latent variables to images in the pixel space. The forward model in latent space is then given by $y = \mathbf{A}G(z) + e$. In

*Table 1. Qualitative and per-sample cost comparison of the sampling methods. Here $C_f$ denotes the cost of one exact forward solve, and we assume that an adjoint solve has the same cost. The quantities $n_{\text{iter}}$ and $n_{\text{CG}}$ denote the number of linear-solver iterations used by `Latent-IMH` and `Proximal-IMH`, respectively, and $n_\ell$ denotes the NUTS tree depth.*

|  | `Approx-IMH` | `Latent-IMH` | `Proximal-IMH` | NUTS |
|---|:---:|:---:|:---:|:---:|
| Nonsquare $\mathbf{F}$ | ✓ | × | ✓ | ✓ |
| Nonlinear $\mathbf{F}$ | ✓ | × | ✓ | ✓ |
| Independent proposal | ✓ | ✓ | ✓ | × |
| Cost per sample | $C_f$ | $n_{\text{iter}}C_f$ | $2n_{\text{CG}}C_f$ | $2^{n_\ell+1}C_f$ |

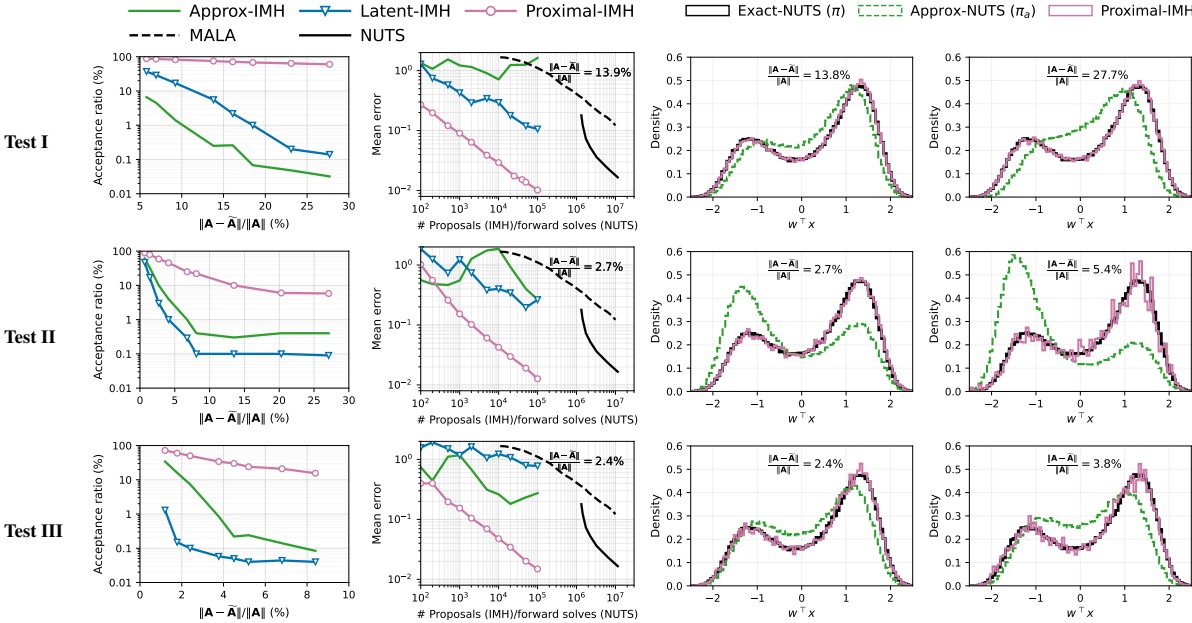

*Figure 2. Comparison of sampling performance in the bimodal test. Rows correspond to Tests I–III. From left to right, panels show Metropolis–Hastings acceptance rates for three IMH methods, convergence of the relative mean error for the IMH methods together with NUTS and MALA, and histograms of the projected samples $w^\top x$ for two levels of operator error. In the histogram panels, Approx-NUTS denotes samples drawn from the `Approx posterior` $\pi_a(x \mid y)$. For `Proximal-IMH`, we set $\beta = \sigma^2$. Results in the left panel are averaged over 5 independent trials for each method.*

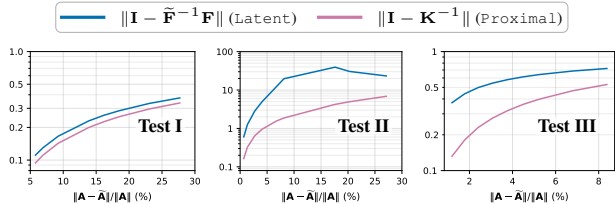

*Figure 3. Operator discrepancy in the bimodal test. The quantities $\|\mathbf{I} - \widetilde{\mathbf{F}}^{-1}\mathbf{F}\|$ and $\|\mathbf{I} - \mathbf{K}^{-1}\|$ are reported for `Latent-IMH` and `Proximal-IMH`, respectively, where $\mathbf{K}$ is defined in Equation (7). For `Proximal-IMH`, the hyperparameter is set to $\beta = \sigma^2$.*

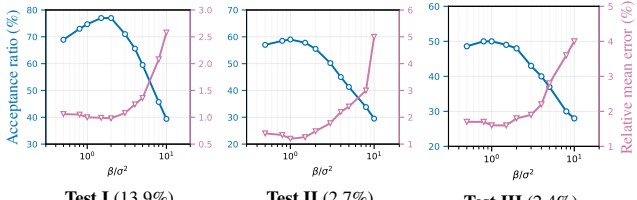

*Figure 4. Effect of $\beta$ on `Proximal-IMH` in the bimodal test. Values in parentheses represent $\|\mathbf{A} - \widetilde{\mathbf{A}}\|/\|\mathbf{A}\|$. Blue curves show acceptance rates (left y-axis), and pink curves show relative mean errors after $10^5$ MH steps (right y-axis).*

our experiments, we set $d_z = 128$, $d_x = 784$, and $d_y = 100$. We focus on sampling the posterior distribution in the latent space:

$$\pi(z \mid y) \propto \exp\left(-\frac{1}{2\sigma^2}\|\mathbf{A}G(z) - y\|^2 - \frac{1}{2}\|z\|^2\right). \quad (22)$$

We randomly select digits from the MNIST test dataset, which were not seen during VAE training, and generate observations through $y = \mathbf{A}x + e$. We consider two types of exact operators $\mathbf{A}$ and approximations $\widetilde{\mathbf{A}}$:

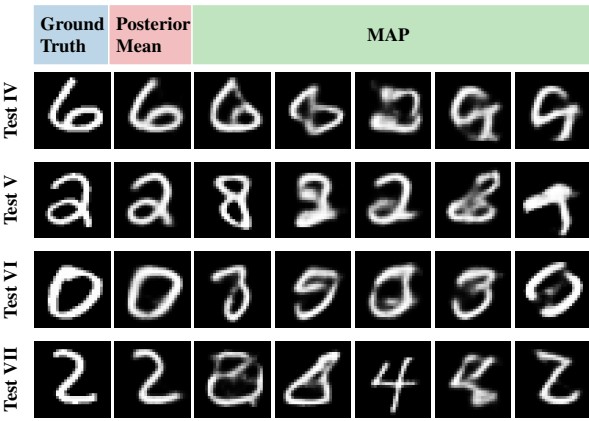

*Figure 5. Ground truth, posterior mean and MAP estimates for the MNIST inverse problem in **Tests IV–VII**. The MAP estimates are obtained from different random initial guesses for the optimization problem. We observe that in several cases the reconstructed MAP point is wrong. This is due to the nonconvexity of solving for a MAP point.*

- **Tests IV and V**: The operators $\mathbf{A}$ and $\widetilde{\mathbf{A}}$ are constructed in the same manner as in Test I. The resulting operator $\mathbf{A}$ has condition number $\kappa(\mathbf{A}) \approx 60$.

- **Tests VI and VII (linearized Helmholtz operator)**: The operator $\mathbf{A}$ is constructed using a linearized Helmholtz operator based on the Born Approximation (Colton & Kress, 1998) on a fine grid ($100 \times 100$). The approximation $\widetilde{\mathbf{A}}$ is constructed on coarser grids ($35 \times 35$ or $20 \times 20$). The operator $\mathbf{A}$ has condition number $\kappa(\mathbf{A}) \approx 420$. Details of the construction procedure are provided in Section C.3.

Figure 5 shows the ground truth, posterior means, and maximum-a-posteriori (MAP) estimates for the MNIST inverse problem. MAP solutions are obtained by minimizing the negative log-posterior in Equation (22) with Adam, using sufficiently many iterations to ensure convergence. The MAP estimates vary substantially across runs, reflecting the nonconvex posterior landscape. In contrast, the posterior means in Figure 6 achieve noticeably smaller reconstruction errors relative to the ground truth.

We report convergence results for `Approx-IMH`, `Proximal-IMH`, and NUTS using relative mean error and relative componentwise second-moment error; `Proximal-IMH` uses a single Gauss–Newton step. Compared with `Approx posterior`, `Proximal-IMH` achieves lower error, faster convergence, reduced variability across trials, and robustness to different observation realizations.

Figure 7 illustrates the effect of $\beta$ on `Proximal-IMH`. Consistent with the bimodal experiments (Figure 4), higher acceptance rates are associated with smaller relative mean errors. In this test, performance is largely insensitive to $\beta$, with relative mean error varying by less than $0.4\%$.

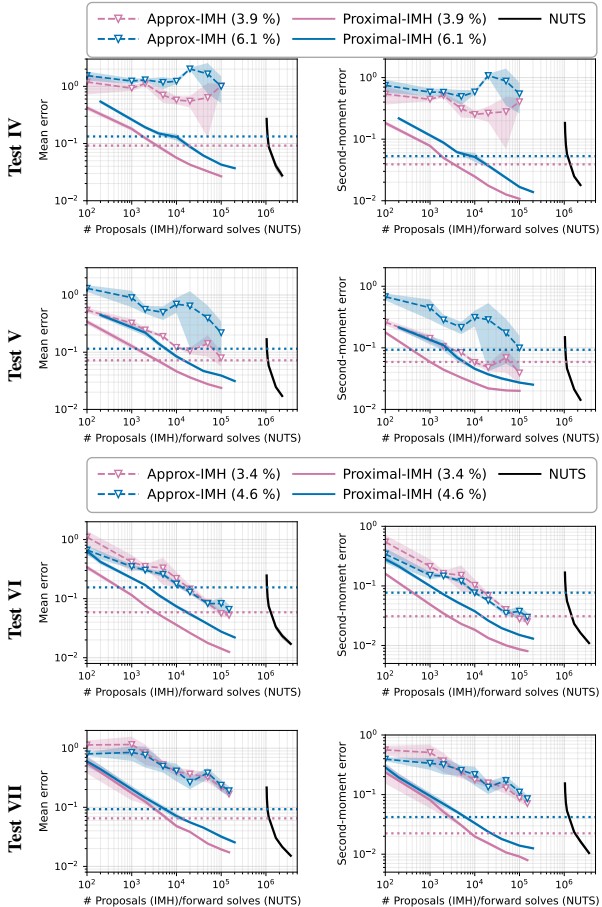

*Figure 6. Sampling performance on MNIST imaging inverse problems: convergence of the relative mean error and relative second-moment error for different sampling methods. Dashed lines indicate the bias induced by the `Approx posterior` $\pi_a(x \mid y)$. For `Proximal-IMH`, we set $\beta = \sigma^2$. Results in the convergence plots are reported over 5 independent trials.*

Figure 8 shows that additional Gauss–Newton correction steps can improve acceptance rates and reduce relative mean error, especially when the relative operator error is larger. However, the gains diminish after two steps, while each extra step increases per-proposal cost, highlighting a cost–accuracy tradeoff.

Consistent with Section 3.4, we monitor the Jacobian determinant ratio for the Gauss–Newton correction; see Figure 13 in Section C.3. The ratio remains close to one for most samples, supporting the approximation used in the nonlinear implementation.

These diagnostics suggest using a short pilot run in nonlinear problems. Starting from $\beta = \sigma^2$, one can tune $\beta$ and the number of Gauss–Newton steps by monitoring the acceptance rate and the proximal objective in Equation (3b). The same pilot run can diagnose whether the Jacobian determinant ratio should be included in the acceptance probability.

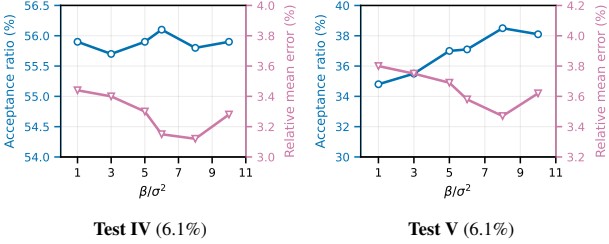

*Figure 7. Effect of the hyperparameter $\beta$ on `Proximal-IMH` in the inverse problem in MNIST. Values in parentheses represent $\|\mathbf{A} - \widetilde{\mathbf{A}}\|/\|\mathbf{A}\|$. Blue curves show acceptance rates (left y-axis), and pink curves show relative mean errors after 10K MH steps (right y-axis). The results are not sensitive to $\beta$.*

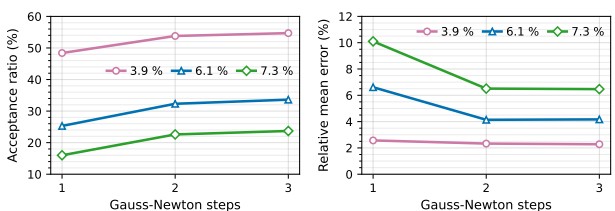

*Figure 8. Effect of the number of Gauss–Newton correction steps in `Proximal-IMH` for Test IV. The labels 3.9%, 6.1%, and 7.3% denote the relative operator error $\|\mathbf{A} - \widetilde{\mathbf{A}}\|/\|\mathbf{A}\|$. Relative mean error is reported after 60K MH steps.*

### 4.3. Nonlinear Helmholtz Test

We consider a nonlinear inverse problem arising from acoustic scattering governed by the Helmholtz equation. The unknown medium $x$ is represented on a $32 \times 32$ grid, while the wavefield is discretized on a finer grid $128 \times 128$. We consider 4 independent source terms. The forward operator $\mathbf{A}_i(x)$ is defined implicitly by solving a Helmholtz equation with homogeneous Neumann boundary conditions (see Section C.4 for details). The Helmholtz operator is nonlinear with respect to the scatterer properties. Assuming independent observations, the negative log-posterior $-\log \pi(x \mid \{y_i\})$ (up to an additive constant) is

$$\frac{1}{2\sigma^2} \sum_{i=1}^{4} \|y_i - \mathbf{A}_i(x)\|^2 + TV_\epsilon(x), \quad (23)$$

where $TV_\epsilon(x)$ denotes a smoothed total-variation prior.

To construct an approximate forward model, we solve the Helmholtz equation on a coarser $64 \times 64$ grid. All linear systems arising in the forward and adjoint solves are computed using GMRES with a Neumann–DCT preconditioner; implementation details are provided in Section C.4.

Figure 9 shows the ground truth, the MAP estimate, and the posterior mean. The MAP estimate is closer to the ground truth because the smoothed TV prior is weakly informative, allowing the likelihood to dominate the posterior; this

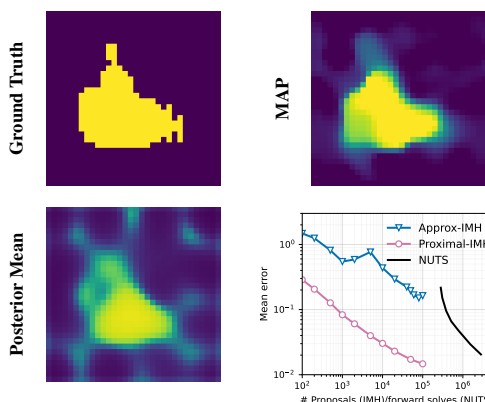

*Figure 9. Nonlinear Helmholtz inverse problem. Ground truth, MAP estimate, and posterior mean for the acoustic scattering test, together with convergence of the relative mean error.*

observation is further corroborated by Figure 15 in the appendix, where the corresponding wavefield solutions $u_i$ for the ground truth, MAP, and posterior mean are nearly indistinguishable. From the convergence results in Figure 9, `Proximal-IMH` consistently outperforms `Approx-IMH` and achieves accuracy comparable to NUTS, while retaining the computational advantages of independent proposals.

## 5. Conclusion

We introduced `Proximal-IMH`, an independence Metropolis–Hastings sampler for inverse problems with approximate forward operators. The method constructs global proposals by correcting samples from an approximate posterior through a stable optimization step, thereby improving their alignment with the exact posterior. For linear forward models, we showed that `Proximal-IMH` achieves smaller expected KL divergence and faster mixing than existing IMH schemes under suitable regularity assumptions. Numerical experiments on multimodal, imaging, and nonlinear PDE-based inverse problems demonstrate improved acceptance rates, faster convergence, and higher sampling accuracy.

**Limitations.** Our theoretical guarantees are derived for linear forward models under regularity assumptions on the prior; extending comparable guarantees to nonlinear forward models is left for future work. The nonlinear extension relies on local Gauss–Newton corrections and, in our implementation, omits the Jacobian determinant ratio, making the nonlinear sampler approximate and potentially biased. Performance also depends on the quality of the approximate operator $\widetilde{\mathbf{F}}$, the regularization parameter $\beta$, and, in nonlinear problems, the number of Gauss–Newton correction steps.

## Acknowledgements

This material is based upon work supported by NSF award OAC 22042261; and Cooperative Agreement 2421782 and the Simons Foundation award MPS-AI-00010515 (NSF-Simons AI Institute for Cosmic Origins—CosmicAI, `https://www.cosmicai.org/`); by the U.S. Department of Energy, Office of Science, Office of Advanced Scientific Computing Research, Applied Mathematics program, Mathematical Multifaceted Integrated Capability Centers (MMICCS) program, under award number DE-SC0023171; by the U.S. Department of Energy, National Nuclear Security Administration Award Number DE-NA0003969; and by the U.S. National Institute on Aging under award number R21AG074276-01. Any opinions, findings, and conclusions or recommendations expressed herein are those of the authors and do not necessarily reflect the views of the DOE, NIH, and NSF. Computing time on the Texas Advanced Computing Centers Stampede system was provided by an allocation from TACC and the NSF.

## Impact Statement

This paper aims to advance the field of Machine Learning. While our work may have various societal impacts, none require specific mention here.

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

# A. Proofs of Main Results

## A.1. Proof of Theorem 3.2

Under the Gaussian setting, the expected KL divergence $\mathbb{D}_a$ admits the following closed-form expression:

$$\mathbb{D}_a = \underbrace{\log \frac{|\boldsymbol{\Sigma}|}{|\boldsymbol{\Sigma}_a|} + \mathrm{Tr}(\boldsymbol{\Sigma}^{-1}\boldsymbol{\Sigma}_a) - d}_{\text{covariance mismatch}} \tag{24}$$

$$+ \underbrace{\frac{1}{\sigma^2}\|\mathbf{A}\boldsymbol{\Delta}_a\mathbf{A}\|_F^2 + \sigma^2\|\boldsymbol{\Delta}_a\|_F^2 + 2\|\boldsymbol{\Delta}_a\mathbf{A}\|_F^2}_{\text{mean mismatch}},$$

where $\boldsymbol{\Delta}_a := \mathbf{A}_a^\dagger - \mathbf{A}^\dagger$. The explicit forms of $\boldsymbol{\Sigma}$, $\boldsymbol{\Sigma}_a$, $\mathbf{A}^\dagger$, and $\mathbf{A}_a^\dagger$ are given in Table 2.

The expressions for $\mathbb{D}_l$ and $\mathbb{D}_p$ follow the same structure as Equation (24), with $(\mathbf{A}_a^\dagger, \boldsymbol{\Sigma}_a)$ replaced by $(\mathbf{A}_l^\dagger, \boldsymbol{\Sigma}_l)$ and $(\mathbf{A}_p^\dagger, \boldsymbol{\Sigma}_p)$, respectively.

*Proof of Theorem 3.2.* The expressions for $\mathbb{D}_a$ and $\mathbb{D}_l$ have been proved in Proposition 3.3 in (Chen & Biros, 2026). We only need to prove the expression for $\mathbb{D}_p$ here.

$$\mathbb{D}_p = \log \frac{|\boldsymbol{\Sigma}|}{|\boldsymbol{\Sigma}_p|} + \mathrm{Tr}(\boldsymbol{\Sigma}^{-1}\boldsymbol{\Sigma}_p) - d + \frac{1}{\sigma^2}\|\mathbf{A}\boldsymbol{\Delta}_p\mathbf{A}\|_F^2 + \sigma^2\|\boldsymbol{\Delta}_p\|_F^2 + 2\|\boldsymbol{\Delta}_p\mathbf{A}\|_F^2, \tag{25}$$

where $\boldsymbol{\Sigma}$ and $\boldsymbol{\Sigma}_p$ are the covariances of `Exact posterior` and `Proximal posterior`, and $\boldsymbol{\Delta}_p := \mathbf{A}_p^\dagger - \mathbf{A}^\dagger$ is the pseudo-inverse difference.

By the diagonal assumptions of $\mathbf{F}$, $\widetilde{\mathbf{F}}$ and $\mathbf{O}$, we have $\mathbf{A}, \widetilde{\mathbf{A}} \in \mathbb{R}^{d_y \times d}$ are matrices with zero elements except for:

$$\mathbf{A}_{ii} = s_i, \qquad \widetilde{\mathbf{A}}_{ii} = \alpha_i s_i, \quad \forall i \in [d_y]. \tag{26}$$

By the formulations listed in Table 2, $\mathbf{A}^\dagger, \mathbf{A}_a^\dagger, \mathbf{A}_p^\dagger \in \mathbb{R}^{d \times d_y}$ have all zero entries except for:

$$\mathbf{A}_{ii}^\dagger = \frac{s_i}{s_i^2 + \sigma^2}, \qquad \left(\mathbf{A}_a^\dagger\right)_{ii} = \frac{\alpha_i s_i}{\alpha_i^2 s_i^2 + \sigma^2}, \qquad \left(\mathbf{A}_p^\dagger\right)_{ii} = \frac{\alpha_i^2 s_i^3 + \sigma^2}{(s_i^2 + \sigma^2)(\alpha_i^2 s_i^2 + \sigma^2)}, \quad \forall i \in [d_y]. \tag{27}$$

Then $\boldsymbol{\Delta}_p \in \mathbb{R}^{d \times d_y}$ is matrix with zero elements except for:

$$(\boldsymbol{\Delta}_p)_{ii} = (\mathbf{A}_p^\dagger)_{ii} - \mathbf{A}_{ii}^\dagger = \frac{\sigma^2 s_i(\alpha_i - 1)}{(\alpha_i^2 s_i^2 + \sigma^2)(s_i^2 + \sigma^2)}, \quad \forall i \in [d_y]. \tag{28}$$

In addition, the covariance of the posterior $\boldsymbol{\Sigma}, \boldsymbol{\Sigma}_p \in \mathbb{R}^{d \times d}$ are diagonal matrices with diagonals as

$$\begin{cases} \boldsymbol{\Sigma}_{ii} = \frac{\sigma^2}{s_i^2 + \sigma^2}, & (\boldsymbol{\Sigma}_p)_{ii} = \frac{\sigma^2 \rho_i}{s_i^2 + \sigma^2}, \quad i \leq d_y, \\ \boldsymbol{\Sigma}_{ii} = 1, & (\boldsymbol{\Sigma}_p)_{ii} = 1, \qquad i > d_y. \end{cases} \tag{29}$$

Thus, for the covariance mismatch of Equation (25), we have

$$\log \frac{|\boldsymbol{\Sigma}|}{|\boldsymbol{\Sigma}_p|} = \sum_{i \in [d_y]} \log \frac{1}{\rho_i}, \quad \mathrm{Tr}(\boldsymbol{\Sigma}^{-1}\boldsymbol{\Sigma}_p) = \sum_{i \in [d_y]} \rho_i + d - d_y. \tag{30}$$

Based on Equation (28), we have

$$(\mathbf{A}\boldsymbol{\Delta}_p)_{ii} = \frac{\sigma^2 s_i^2(\alpha_i - 1)}{(\alpha_i^2 s_i^2 + \sigma^2)(s_i^2 + \sigma^2)} \tag{31}$$

Define $\psi_i = (\alpha_i^2 s_i^2 + \sigma^2)(s_i^2 + \sigma^2)$, we have formulations for the mean mismatch terms in Equation (25):

$$\frac{1}{\sigma^2}\|\mathbf{A}\boldsymbol{\Delta}_p\mathbf{A}\|_F^2 = \sum_{i \in [d_y]} \frac{\sigma^2 s_i^6(\alpha_i - 1)^2}{\psi_i^2}, \quad \sigma^2\|\boldsymbol{\Delta}_p\|_F^2 = \sum_{i \in [d_y]} \frac{\sigma^6 s_i^2(\alpha_i - 1)^2}{\psi_i^2}, \quad 2\|\boldsymbol{\Delta}_p\mathbf{A}\|_F^2 = \sum_{i \in [d_y]} \frac{2\sigma^4 s_i^4(\alpha_i - 1)^2}{\psi_i^2}. \tag{32}$$

Thus, the mean mismatch term is

$$\frac{1}{\sigma^2}\|\mathbf{A}\boldsymbol{\Delta}_p\mathbf{A}\|_F^2 + \sigma^2\|\boldsymbol{\Delta}_p\|_F^2 + 2\|\boldsymbol{\Delta}_p\mathbf{A}\|_F^2 = \sum_{i\in[d_y]}\frac{\sigma^2 s_i^2(\alpha_i-1)^2}{(\alpha_i^2 s_i^2+\sigma^2)^2} = \sum_{i\in[d_y]}\zeta_i(\alpha_i-1)^2 s_i^2\sigma^2. \tag{33}$$

Substituting Equations (30) and (33) into Equation (25), we can get the expression for $\mathbb{D}_p$ in Theorem 3.2. $\qquad\square$

## A.2. Mixing Time

**Assumption A.1.** The exact posterior $\pi(x|y)$ is strongly log-concave, i.e., $-\log\pi(x|y)$ is $m$-strongly convex on $\mathbb{R}^d$.

**Assumption A.2.** Log-weight functions are locally Lipschitz. For any $R > 0$, there exist constants $C_a(R) \geq 0$ such that for all $(x, x') \in \mathsf{Ball}(x^*, R)$, $\|w_a(x) - w_a(x')\| \leq C_a(R)\|x - x'\|$.

For the functions $w_l(x)$ and $w_p(x)$, we assume that this condition holds with constants $C_l(R)$ and $C_p(R)$, respectively.

**Theorem A.3** (Adapted from Theorem 4.3 in (Chen & Biros, 2026)). *Let $\epsilon \in (0, 1)$. Assume that $\pi_a(x \mid y)$, $\pi_l(x \mid y)$ and $\pi_p(x \mid y)$ are all $\gamma$-warm starts with respect to the exact posterior $\pi(x|y)$ (i.e., for any Borel set $\mathcal{E}$, $\mu_0(\mathcal{E}) \leq \gamma\pi(\mathcal{E})$). Assume that Assumption A.1 and Assumption A.2 hold. For Assumption A.2, suppose that the Lipschitz conditions hold with $C_a(R_a)$, $C_l(R_l)$ and $C_p(R_p) \leq \log 2\sqrt{m}/32$, where*

$$R_a = \max\left\{\sqrt{\frac{d}{m}}r\left(\frac{\epsilon}{17\gamma}\right), \sqrt{\frac{d}{m}}r\left(\frac{\epsilon}{272\gamma}\right) + \|x^* - x_a^*\|\right\}. \tag{34}$$

$$R_l = \max\left\{\sqrt{\frac{d}{m}}r\left(\frac{\epsilon}{17\gamma}\right), \quad \sqrt{\frac{d}{m}}r\left(\frac{\epsilon}{272\gamma}\right)\frac{1}{\sigma_{\min}(\widetilde{\mathbf{F}}^{-1}\mathbf{F})} + \|x^* - x_l^*\|\right\}, \tag{35}$$

$$R_p = \max\left\{\sqrt{\frac{d}{m}}r\left(\frac{\epsilon}{17\gamma}\right), \quad \sqrt{\frac{d}{m}}r\left(\frac{\epsilon}{272\gamma}\right)\sigma_{\max}(\mathbf{K}) + \|x^* - x_p^*\|\right\}, \tag{36}$$

*where $\sigma_{\min}(\cdot)$ and $\sigma_{\max}(\cdot)$ are the smallest and largest singular values of the matrix, $r(\cdot)$ is a constant defined by*

$$r(s) = 2 + 2\max\left\{\frac{-\log^{1/4}(s)}{d^{1/4}}, \frac{-\log^{1/2}(s)}{d^{1/2}}\right\}. \tag{37}$$

*Then the mixing time for* `Approx-IMH` *has*

$$\tau_{mix}^a(\epsilon) \leq 128\log\left(\frac{2\gamma}{\epsilon}\right)\max\left(1, \frac{128^2 C_a^2(R_a)}{(\log 2)^2 m}\right). \tag{38}$$

*The mixing time for* `Latent-IMH` *and* `Proximal-IMH`, *$\tau_{mix}^l(\epsilon)$ and $\tau_{mix}^p(\epsilon)$ satisfy a similar bound with $C_a(R_a)$ replaced by $C_l(R_l)$ and $C_p(R_p)$ respectively.*

*Proof.* The results for `Approx-IMH` and `Latent-IMH` were established in Theorem 4.3 of (Chen & Biros, 2026). We follow a similar argument to the proof of `Approx-IMH` to establish the result for `Proximal-IMH`.

Since

$$\pi_p(x \mid y) \propto q\left(y - \widetilde{\mathbf{A}}\mathbf{K}^{-1}x\right)p(\mathbf{K}^{-1}x),$$

Hence $\pi_p(x \mid y)$ is $m\,\sigma_{\min}^2(\mathbf{K}^{-1})$-strongly log-concave. Consequently, there exists a constant $c_2 > 0$ such that

$$\pi_p\big(\mathsf{Ball}(x_p^*, R_2)\big) \geq 1 - c_2 s,$$

where

$$R_2 = \sqrt{\frac{d}{m}}\,\sigma_{\max}(\mathbf{K})\,r(c_2 s). \tag{39}$$

If $C_p(R_p) \leq \frac{\log(2\sqrt{m})}{32}$, where $R_p$ satisfies Equation (36), then the conditions of Equation (96) in (Chen & Biros, 2026) are satisfied. The desired mixing-time bound for `Proximal-IMH` therefore follows. $\qquad\square$

*Proof of Theorem 3.3.* We prove the result by invoking Theorem A.3. Our goal is to derive local Lipschitz constants $C_a(R_a)$, $C_l(R_l)$, and $C_p(R_p)$ for log-weight functions $w_a(x)$, $w_l(x)$, and $w_p(x)$, respectively. Without loss of generality, we assume that the posterior mode is

$$x^\star := \arg\min_x \; \pi(x \mid y)$$

is located at the origin, i.e., $x^\star = 0$. Throughout, we denote by $B(R)$ the Euclidean ball centered at the origin with radius $R > 0$.

1. `Approx-IMH`: The log-weight function is

$$w_a(x) = \log \frac{\pi(x|y)}{\pi_a(x|y)} = -\frac{1}{2\sigma^2}\left(\|y - \mathbf{A}x\|^2 - \|y - \widetilde{\mathbf{A}}x\|^2\right)$$

$$= \frac{1}{2\sigma^2}\left(x^\top \underbrace{(\mathbf{A} + \widetilde{\mathbf{A}})^\top \Delta\mathbf{A}}_{=:\mathbf{M}_a} x + 2y^\top \Delta\mathbf{A}x\right). \tag{40}$$

Define the symmetric version of $\mathbf{M}_a$ by $\mathbf{M}_{a,s} = \frac{1}{2}\left(\mathbf{M}_a + \mathbf{M}_a^\top\right)$, then $x^\top \mathbf{M}_a x = x^\top \mathbf{M}_{a,s} x$. Thus for any $x_1, x_2 \in B(R_a)$

$$\left|x_1^\top \mathbf{M}_{a,s}x_1 - x_2^\top \mathbf{M}_{a,s}x_2\right| = \left|(x_1 + x_2)^\top \mathbf{M}_{a,s}(x_1 - x_2)\right| \leq \|\mathbf{M}_{a,s}\|\|x_1 + x_2\|\|x_1 - x_2\|$$

$$\leq 4R_a\|\mathbf{A}\|\|\Delta\mathbf{A}\|\|x_1 - x_2\|. \tag{41}$$

Besides, we have

$$\left|2y^\top \Delta\mathbf{A}x_1 - 2y^\top \Delta\mathbf{A}x_2\right| \leq 2\|\Delta\mathbf{A}^\top y\|\|x_1 - x_2\| \leq 2\|y\|\|\Delta\mathbf{A}\|\|x_1 - x_2\|. \tag{42}$$

Combining Equations (40) to (42), we have

$$|w_a(x_1) - w_a(x_2)| \leq \underbrace{\frac{2R_a\|\mathbf{A}\| + \|y\|}{\sigma^2}\|\Delta\mathbf{A}\|}_{=:C_a(R_a)}\|x_1 - x_2\|. \tag{43}$$

When $d$ is large, by Theorem A.3, $R_a \sim \sqrt{\frac{d}{m}}$, thus

$$C_a(R_a) \sim \sqrt{\frac{d}{m}}\frac{\|\mathbf{A}\|}{\sigma^2}\|\Delta\mathbf{A}\|,$$

$$\tau_{mix}^a \sim \frac{d}{m^2}\frac{\|\mathbf{A}\|^2}{\sigma^4}\|\Delta\mathbf{A}\|^2. \tag{44}$$

2. `Latent-IMH`: Denote $f$ as the negative log prior, i.e. $f(x) = -\log p(x)$, then the log-weight function is

$$w_l(x) = \log \frac{\pi(x|y)}{\pi_l(x|y)} = -\left(f(x) - f(\mathbf{M}_l x)\right), \qquad \mathbf{M}_l := \widetilde{\mathbf{F}}^{-1}\mathbf{F}. \tag{45}$$

Thus we have gradient of the log-weight function

$$\nabla w_l(x) = \nabla f(x) - \mathbf{M}_l^\top \nabla f(\mathbf{M}_l x)$$

$$= \left(\mathbf{I} - \mathbf{M}_l^\top\right)\nabla f(x) + \mathbf{M}_l^\top\left(\nabla f(x) - \nabla f(\mathbf{M}_l x)\right) \tag{46}$$

By assumption in Theorem 3.3, $f$ is $L-$smooth, thus

$$\|\nabla w_l(x)\| \leq \|\mathbf{I} - \mathbf{M}_l\|\|\nabla f(x)\| + \|\mathbf{M}_l\|\|\nabla f(x) - \nabla f(\mathbf{M}_l x)\|$$

$$\leq \|\mathbf{I} - \mathbf{M}_l\|\|\nabla f(x)\| + \|\mathbf{M}_l\|L\|x - \mathbf{M}_l x\|$$

$$\leq \|\mathbf{I} - \mathbf{M}_l\|\left(\|\nabla f(x)\| + L\|\mathbf{M}_l\|\|x\|\right) \tag{47}$$

For any $x_1, x_2 \in B(R_l)$, we have

$$|w_l(x_1) - w_l(x_2)| \leq \sup_{x \in B(R_l)} \|\nabla w_l(x)\| \|x_1 - x_2\|. \tag{48}$$

By Equation (47), we have

$$\sup_{x \in B(R_l)} \|\nabla w_l(x)\| \leq \|\mathbf{I} - \mathbf{M}_l\| \left( \sup_{x \in B(R_l)} \|\nabla f(x)\| + LR_l \|\mathbf{M}_l\| \right), \tag{49}$$

where

$$\sup_{x \in B(R_l)} \|\nabla f(x)\| \leq \|\nabla f(0)\| + L\|x\| \leq \|\nabla f(0)\| + LR_l. \tag{50}$$

Substitute into Equation (48):

$$|w_l(x_1) - w_l(x_2)| \leq \underbrace{\|\mathbf{I} - \mathbf{M}_l\| \left[ (1 + \|\mathbf{M}_l\|) LR_l + \|\nabla f(0)\| \right]}_{=:C_l(R_l)} \|x_1 - x_2\|. \tag{51}$$

When $d$ is large, by Theorem A.3, $R_l \sim \sqrt{\frac{d}{m}}$, thus

$$C_l(R_l) \sim L\sqrt{\frac{d}{m}} \left( 1 + \|\mathbf{M}_l\| \right) \|\mathbf{I} - \mathbf{M}_l\|,$$

$$\tau_{mix}^l \sim \frac{L^2 d}{m^2} \left( 1 + \|\mathbf{M}_l\| \right)^2 \|\mathbf{I} - \mathbf{M}_l\|^2. \tag{52}$$

3. `Proximal-IMH`: Denote $f(x) = -\log p(x)$, by Equation (5), we have `Proximal posterior`:

$$\pi_p(x|y) \propto \exp\left( -\frac{1}{2\sigma^2} \|\widetilde{\mathbf{A}}\mathbf{K}^{-1}x - y\|^2 - f(\mathbf{K}^{-1}x) \right) \tag{53}$$

the log-weight function is

$$w_p(x) = \log \frac{\pi(x|y)}{\pi_p(x|y)} = -\frac{1}{2\sigma^2} \underbrace{\left( \|\mathbf{A}x - y\|^2 - \|\widetilde{\mathbf{A}}\mathbf{K}^{-1}x - y\|^2 \right)}_{=:h(x)} - \left( f(x) - f(\mathbf{K}^{-1}x) \right). \tag{54}$$

According to Section 3.1, given $\widetilde{x}$, define $x = \mathbf{K}\widetilde{x}$. Then $x$ is the solution to the following optimization:

$$\min_x \|\mathbf{A}x - \widetilde{\mathbf{A}}\widetilde{x}\|^2 + \beta\|x - \widetilde{x}\|^2. \tag{55}$$

Taking gradient of the objective, we can get:

$$\mathbf{A}^\top r = -\beta(x - \widetilde{x}), \qquad r = \mathbf{A}x - \widetilde{\mathbf{A}}\widetilde{x}. \tag{56}$$

Thus

$$
\begin{aligned}
h(x) = h(\mathbf{K}\widetilde{x}) &= \|\mathbf{A}\mathbf{K}\widetilde{x} - y\|^2 - \left\|\widetilde{\mathbf{A}}\widetilde{x} - y\right\|^2 \\
&= \left( \mathbf{A}\mathbf{K}\widetilde{x} + \widetilde{\mathbf{A}}\widetilde{x} - 2y \right)^\top \left( \mathbf{A}\mathbf{K}\widetilde{x} - \widetilde{\mathbf{A}}\widetilde{x} \right) = (\mathbf{A}\mathbf{K}\widetilde{x} + \mathbf{A}\widetilde{x} + \Delta\mathbf{A}\widetilde{x} - 2y)^\top (\mathbf{A}x - \widetilde{\mathbf{A}}\widetilde{x}) \\
&\overset{\text{Equation (56)}}{=} -\beta(x + \widetilde{x})^\top(x - \widetilde{x}) - 2y^\top r + \widetilde{x}^\top(\Delta\mathbf{A})^\top r.
\end{aligned}
\tag{57}
$$

For $y$, we have $y$ generated by some $x_0$ by $y = \mathbf{A}x_0 + e$, where $e$ is the noise, we neglect the noise term because we discuss the scaling of the mixing time. Thus by Equation (56)

$$y^\top r = x_0^\top \mathbf{A}^\top r = -\beta x_0^\top (x - \widetilde{x}). \tag{58}$$

Let $\mathbf{A}^{\dagger} = (\mathbf{A}^{\top}\mathbf{A} + \beta\mathbf{I})^{-1}\mathbf{A}^{\top}$, we have $\mathbf{K} = \mathbf{I} + \mathbf{A}^{\dagger}\Delta\mathbf{A}$. For the last term in Equation (57), we have

$$
\begin{aligned}
\widetilde{x}^{\top}(\Delta\mathbf{A})^{\top}r &= \widetilde{x}^{\top}(\Delta\mathbf{A})^{\top}\left(\mathbf{A}\mathbf{K}\widetilde{x} - \widetilde{\mathbf{A}}\widetilde{x}\right) = \widetilde{x}^{\top}(\Delta\mathbf{A})^{\top}\left(\mathbf{A}\left(\mathbf{I} + \mathbf{A}^{\dagger}\Delta\mathbf{A}\right) - \widetilde{\mathbf{A}}\right)\widetilde{x} \\
&= \widetilde{x}^{\top}(\Delta\mathbf{A})^{\top}\left(\mathbf{A}\mathbf{A}^{\dagger} - \mathbf{I}\right)(\Delta\mathbf{A})\widetilde{x}.
\end{aligned}
\tag{59}
$$

Substituting Equations (58) and (59) into Equation (57), we can get

$$
\begin{aligned}
h(x) &= -\beta(x + \mathbf{K}^{-1}x)^{\top}(x - \mathbf{K}^{-1}x) + 2\beta x_0^{\top}(x - \mathbf{K}^{-1}x) + (\mathbf{K}^{-1}x)^{\top}(\Delta\mathbf{A})^{\top}\left(\mathbf{A}\mathbf{A}^{\dagger} - \mathbf{I}\right)(\Delta\mathbf{A})(\mathbf{K}^{-1}x) \\
&= -\beta x^{\top}\left(\mathbf{I} + \mathbf{K}^{-1}\right)^{\top}\left(\mathbf{I} - \mathbf{K}^{-1}\right)x - 2\beta x_0^{\top}(\mathbf{I} - \mathbf{K}^{-1})x + x^{\top}\left(\mathbf{K}^{-T}(\Delta\mathbf{A})^{\top}\left(\mathbf{A}\mathbf{A}^{\dagger} - \mathbf{I}\right)(\Delta\mathbf{A})\mathbf{K}^{-1}\right)x.
\end{aligned}
\tag{60}
$$

Thus, for any $x_1, x_2 \in B(R_p)$, we have

$$
\begin{aligned}
\frac{1}{2\sigma^2}\left|h(x_1) - h(x_2)\right| &\leq \frac{\beta}{2\sigma^2}\left\|\mathbf{I} + \mathbf{K}^{-1}\right\|\left\|\mathbf{I} - \mathbf{K}^{-1}\right\|\|x_1 + x_2\|\|x_1 - x_2\| \\
&\quad + \frac{\beta}{\sigma^2}\left\|\mathbf{I} - \mathbf{K}^{-1}\right\|\|x_0\|\|x_1 - x_2\| + \frac{\|\mathbf{A}\mathbf{A}^{\dagger} - \mathbf{I}\|}{\sigma^2}\|\mathbf{K}^{-1}\|^2\|\Delta\mathbf{A}\|^2\|x_1 + x_2\|\|x_1 - x_2\| \\
&\lesssim R_p\left\|\mathbf{I} + \mathbf{K}^{-1}\right\|\left\|\mathbf{I} - \mathbf{K}^{-1}\right\|\|x_1 - x_2\|,
\end{aligned}
\tag{61}
$$

where we derived the last equation by using $\beta = \Theta(\sigma^2)$.

As for the local Lipschitzness of another term in Equation (54), we adopt what we derived for the `Latent-IMH`, i.e. by (51), we have for any $x_1, x_2 \in B(R_p)$,

$$
\left|\left(f(x_1) - f(\mathbf{K}^{-1}x_1)\right) - \left(f(x_2) - f(\mathbf{K}^{-1}x_2)\right)\right| \leq \left\|\mathbf{I} - \mathbf{K}^{-1}\right\|\left[(1 + \|\mathbf{K}^{-1}\|)LR_p + \|\nabla f(0)\|\right]\|x_1 - x_2\|.
\tag{62}
$$

Combining Equations (54), (61) and (62), we have for any $x_1, x_2 \in B(R_p)$

$$
|w_p(x_1) - w_p(x_2)| \lesssim C_p(R_p)\|x_1 - x_2\|,
\tag{63}
$$

where

$$
C_p(R_p) = (L + 1)R_p(1 + \|\mathbf{K}^{-1}\|)\left\|\mathbf{I} - \mathbf{K}^{-1}\right\|.
\tag{64}
$$

When $d$ is large, by Theorem A.3, $R_p \sim \sqrt{\frac{d}{m}}$, and

$$
\tau_{mix}^p \sim \frac{d(L + 1)^2}{m^2}(1 + \|\mathbf{K}^{-1}\|)^2\left\|\mathbf{I} - \mathbf{K}^{-1}\right\|^2.
\tag{65}
$$

$\square$

## A.3. Derivation of Equation (20)

Let $\widetilde{\mathbf{A}}G(x)$ denote an approximate forward model and assume that $\Delta\mathbf{A} = \delta\mathbf{A}$ for some $\delta \in (0, 1)$. The residual is $r(\widetilde{x}) = \mathbf{A}G(\widetilde{x}) - \widetilde{\mathbf{A}}G(\widetilde{x}) = \delta\mathbf{A}\,G(\widetilde{x})$. Then the Gauss–Newton update from Equation (17) becomes

$$
GN(\widetilde{x}) := \widetilde{x} - \delta\left(\mathbf{J}(\widetilde{x})^{\top}\mathbf{J}(\widetilde{x}) + \beta\mathbf{I}\right)^{-1}\mathbf{J}(\widetilde{x})^{\top}\mathbf{A}G(\widetilde{x}),
\tag{66}
$$

For small residuals $r$, we neglect second-order terms in the linearization of the Gauss–Newton update Equation (17). Under this approximation, the Jacobian of a single Gauss–Newton step Equation (66) takes the form

$$
\begin{aligned}
\mathbf{J}_{GN}(\widetilde{x}) &= \mathbf{I} - \delta\left(\mathbf{J}(\widetilde{x})^{\top}\mathbf{J}(\widetilde{x}) + \beta\mathbf{I}\right)^{-1}\mathbf{J}(\widetilde{x})^{\top}(\mathbf{A}\mathbf{J}_G(\widetilde{x})) \\
&= \mathbf{I} - \delta\left(\mathbf{J}(\widetilde{x})^{\top}\mathbf{J}(\widetilde{x}) + \beta\mathbf{I}\right)^{-1}\mathbf{J}(\widetilde{x})^{\top}\mathbf{J}(\widetilde{x}) \\
&= (1 - \delta)\mathbf{I} + \delta\beta\left(\mathbf{J}^{\top}(\widetilde{x})\mathbf{J}(\widetilde{x}) + \beta\mathbf{I}\right)^{-1},
\end{aligned}
\tag{67}
$$

where the second equality is derived by the fact that $\mathbf{J}(\widetilde{x}) = \mathbf{A}\mathbf{J}_G(\widetilde{x})$.

For two arbitrary points $\widetilde{x}_1$ and $\widetilde{x}_2$, let $\{\lambda_{1,i}\}_{i\in[d_x]}$ and $\{\lambda_{2,i}\}_{i\in[d_x]}$ denote the eigenvalues (in descending order) of $\mathbf{J}^\top(\widetilde{x}_1)\mathbf{J}(\widetilde{x}_1)$ and $\mathbf{J}^\top(\widetilde{x}_2)\mathbf{J}(\widetilde{x}_2)$, respectively. The logarithm of the Jacobian determinant ratio can then be expressed as

$$\log \frac{|\det \mathbf{J}_{GN}(\widetilde{x}_1)|}{|\det \mathbf{J}_{GN}(\widetilde{x}_2)|} = \sum_{i\in[d_x]} \log \frac{1 - \delta\lambda_{1,i}/(\lambda_{1,i} + \beta)}{1 - \delta\lambda_{2,i}/(\lambda_{2,i} + \beta)}. \tag{68}$$

## B. Helmholtz Forward Model

### B.1. Linearized Helmholtz (Born Approximation) Model

Consider the Helmholtz equation

$$\mathcal{L}(x)u := \left(-\Delta - k^2 x\right) u = f, \tag{69}$$

where $x$ denotes the spatially varying contrast parameter and $k > 0$ is the wavenumber. Let $x_0$ be a known background medium and write

$$x = x_0 + \delta x,$$

where $\delta x$ is a small perturbation.

Let $u_0$ denote the background wavefield solving

$$\mathcal{L}(x_0)u_0 = f. \tag{70}$$

Substituting this decomposition into the Helmholtz equation and retaining only first-order terms in $\delta x$ yields the linearized equation

$$\mathcal{L}(x_0)\,\delta u = k^2\,\delta x\,u_0, \tag{71}$$

where $\delta u := u - u_0$ denotes the scattered field.

The scattered field can therefore be expressed as

$$\delta u = \mathcal{L}(x_0)^{-1}\left(k^2\,\delta x\,u_0\right). \tag{72}$$

This relation defines a linear map from the contrast perturbation $\delta x$ to the scattered field $\delta u$. For convenience, we introduce the operator

$$F := \mathcal{L}(x_0)^{-1}\left(k^2\,\mathrm{diag}(u_0)\right), \tag{73}$$

so that $\delta u = F\,\delta x$.

### B.2. Nonlinear Forward Model and Adjoint

For the full nonlinear problem, the forward model is given by

$$\begin{cases} \mathcal{L}(x)u = f, & \text{where } \mathcal{L}(x) := -\Delta - k^2 x \\ y = \mathbf{O}u, \end{cases} \tag{74}$$

where the wavefield $u = u(x)$ depends nonlinearly on the contrast parameter $x$.

Given a current parameter $x$, the forward wavefield $u$ is computed by solving the Helmholtz equation using GMRES in a matrix-free fashion, without explicitly forming the system matrix.

**Adjoint formulation.** Let $r := \mathbf{O}u - y$ denote the data residual. The adjoint field $v$ is defined as the solution of the adjoint Helmholtz equation

$$\mathcal{L}(x)^\top v = \mathbf{O}^\top r, \tag{75}$$

*Table 2. Distributions of* `Exact`*,* `Approx`*,* `Latent` *and* `Proximal` *posteriors for Gaussian prior* $p(x) = \mathcal{N}(\mathbf{0}, \mathbf{I})$ *and noise* $q(e) = \mathcal{N}(\mathbf{0}, \sigma^2 \mathbf{I})$*. For* `Proximal posterior`*,* $\mathbf{K} = (\mathbf{A}^\top \mathbf{A} + \beta \mathbf{I})^{-1}(\mathbf{A}^\top \widetilde{\mathbf{A}} + \beta \mathbf{I})$*.*

| Posterior | Distribution | Mean | Covariance | Pseudo-inverse |
|---|---|---|---|---|
| `Exact` | $\pi(x \mid y) = \mathcal{N}(\mu, \boldsymbol{\Sigma})$ | $\mu = \mathbf{A}^\dagger y$ | $\boldsymbol{\Sigma} = \mathbf{I} - \mathbf{A}^\dagger \mathbf{A}$ | $\mathbf{A}^\dagger = \mathbf{A}^\top \left( \mathbf{A}\mathbf{A}^\top + \sigma^2 \mathbf{I} \right)^{-1}$ |
| `Approx` | $\pi_a(x \mid y) = \mathcal{N}(\mu_a, \boldsymbol{\Sigma}_a)$ | $\mu_a = \mathbf{A}_a^\dagger y$ | $\boldsymbol{\Sigma}_a = \mathbf{I} - \mathbf{A}_a^\dagger \widetilde{\mathbf{A}}$ | $\mathbf{A}_a^\dagger = \widetilde{\mathbf{A}}^\top \left( \widetilde{\mathbf{A}}\widetilde{\mathbf{A}}^\top + \sigma^2 \mathbf{I} \right)^{-1}$ |
| `Latent` | $\pi_l(x \mid y) = \mathcal{N}(\mu_l, \boldsymbol{\Sigma}_l)$ | $\mu_l = \mathbf{A}_l^\dagger y$ | $\boldsymbol{\Sigma}_l = \mathbf{F}^{-1} \widetilde{\mathbf{F}} \boldsymbol{\Sigma}_a \widetilde{\mathbf{F}}^\top \mathbf{F}^{-\top}$ | $\mathbf{A}_l^\dagger = \mathbf{F}^{-1} \widetilde{\mathbf{F}} \mathbf{A}_a^\dagger$ |
| `Proximal` | $\pi_p(x \mid y) = \mathcal{N}(\mu_p, \boldsymbol{\Sigma}_p)$ | $\mu_p = \mathbf{A}_p^\dagger y$ | $\boldsymbol{\Sigma}_p = \mathbf{K}\boldsymbol{\Sigma}_a\mathbf{K}^\top$ | $\mathbf{A}_p^\dagger = \mathbf{K}\mathbf{A}_a^\dagger$ |

| Effects | $\log_{10}$ SNR | $\frac{\|\mathbf{F}-\widetilde{\mathbf{F}}\|}{\|\mathbf{F}\|}$ | $d_y/d$ | $d$ |
|---|---|---|---|---|
| Noise level | 0.5 —4.0 | 6% | 0.2 | 500 |
| Operator error | 2.5 | 2% —21% | 0.2 | 500 |
| Observation ratio | 2.5 | 6% | 0.05 —0.5 | 500 |
| Dimension | 2.5 | 6% | 0.2 | 100 —2000 |

*Table 3. Experimental setup for comparing* $\mathbb{D}_a$*,* $\mathbb{D}_l$ *and* $\mathbb{D}_p$*.*

with adjoint boundary conditions consistent with the forward problem.

The gradient of the data misfit functional

$$\Phi(x) := \tfrac{1}{2}\|\mathbf{O}u - y\|^2 \tag{76}$$

with respect to $x$ is given by

$$\nabla_x \Phi(x) = -k^2\, u \odot v, \tag{77}$$

where $\odot$ is the pointwise (Hadamard) product.

## C. Additional Experimental Details

### C.1. Proposal Comparisons in Section 3.2

In Table 2, we provide the Gaussian formulation of different posteriors when prior is normal and noise is Gaussian distribution with variance $\sigma^2$. Table 3 presents the experimental setup for the sensitivity test on KL divergence in Figure 1.

### C.2. Bimodal Test in Section 4.1

**Histogram for prior samples.** The histogram of the projection of prior samples in the bimodal test (see Equation (21)) is plotted in Figure 10.

**Cost–accuracy comparison.** Figure 11 replots the bimodal-test relative mean error against the number of exact forward/adjoint solves. For `Latent-IMH` and `Proximal-IMH`, the counts include the exact forward/adjoint solves used in the corresponding linear solves. We use unpreconditioned GMRES for `Latent-IMH` and unpreconditioned CG for `Proximal-IMH` in this comparison, making the reported costs conservative for both methods; effective preconditioning can further reduce the number of exact forward/adjoint solves.

### C.3. MNIST Inverse Problem in Section 4.2

**A and $\widetilde{\mathbf{A}}$ construction in Tests VI and VII (Section 4.2).** We consider the linearized Helmholtz forward model introduced in Section B.1. In our experiments, the parameter $x$ is represented on a $28 \times 28$ grid corresponding to MNIST images, while the wavefield $u$ is discretized on a finer $100 \times 100$ spatial grid. Consequently, the dimensions of the parameter and wavefield are $d_x = 784$ and $d_u = 10^4$, respectively. The wave number is set to $k = 2.4\pi$ in our experiments.

The linearized operator $F$ defined in Equation (73) is discretized on a two-dimensional uniform grid using a second-order finite-difference scheme with homogeneous Neumann boundary conditions; we denote the fine-grid inverse Helmholtz

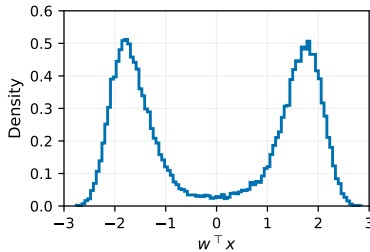

*Figure 10. Histogram of the projection of prior samples in the bimodal test (see Equation (21)).*

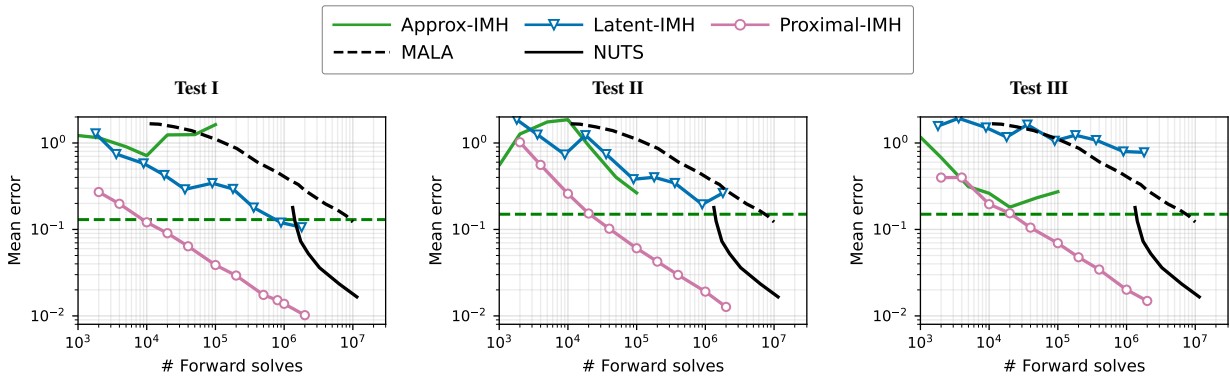

*Figure 11. Cost–accuracy comparison in the bimodal test using the number of exact forward/adjoint solves. The three panels correspond to Tests I–III. The dashed green horizontal line indicates the bias of the* Approx *posterior. For* Latent-IMH *and* Proximal-IMH, *the counts include the exact forward/adjoint solves used in the corresponding linear solves. We use unpreconditioned GMRES for* Latent-IMH *and unpreconditioned CG for* Proximal-IMH, *so this comparison is conservative for both methods; effective preconditioning can further reduce their cost.*

operator by $\mathbf{F} \in \mathbb{R}^{d_u \times d_u}$. A sparse prolongation operator $\mathbf{P} \in \mathbb{R}^{d_u \times d_x}$ maps the parameter $x$ from the coarse $28 \times 28$ grid to the fine $100 \times 100$ grid on which the Helmholtz equation is solved. With an observation operator $\mathbf{O} \in \mathbb{R}^{d_y \times d_u}$, the resulting fine-scale forward operator $\mathbf{A} \in \mathbb{R}^{d_y \times d_x}$ is defined as

$$\mathbf{A} := \mathbf{O}\,\mathbf{F}\,\mathbf{P}. \tag{78}$$

The corresponding forward model from the latent variable $z$ to the observation $y$ is then given by

$$y = \mathbf{A}\,G(z) + e, \tag{79}$$

where $G : \mathbb{R}^{d_z} \to \mathbb{R}^{d_x}$ denotes the decoder of the variational autoencoder (VAE).

To construct the approximate operator $\widetilde{\mathbf{A}}$, we introduce coarser spatial grids of size $\widetilde{n}_u \times \widetilde{n}_u$ for the wavefield $u$, where $\widetilde{n}_u \in \{20, 35\}$ in our experiments. We denote the corresponding discretized Helmholtz operator on the coarse grid by $\widetilde{\mathbf{F}} \in \mathbb{R}^{\widetilde{d}_u \times \widetilde{d}_u}$, with $\widetilde{d}_u = \widetilde{n}_u^2$. Sparse prolongation operators $\mathbf{P}_u \in \mathbb{R}^{d_u \times \widetilde{d}_u}$ and $\mathbf{P}_x \in \mathbb{R}^{\widetilde{d}_u \times d_x}$ are used to map vectors between the coarse and fine grids. The resulting approximate forward operator is then defined as

$$\widetilde{\mathbf{A}} := \mathbf{O}\,\mathbf{P}_u\,\widetilde{\mathbf{F}}\,\mathbf{P}_x. \tag{80}$$

The dimensions involved in the construction of $\mathbf{A}$ and $\widetilde{\mathbf{A}}$ are summarized in Table 4, and schematic diagrams of the corresponding matrix factorizations are shown in Figure 12.

**Jacobian Determinant Diagnostics of the Gauss–Newton Correction.**   Figure 13 examines the variability of the Jacobian determinant associated with a single Gauss–Newton correction across posterior samples and operator approximation levels. In the left panels, we plot the sorted log determinants of the Gauss–Newton Jacobian evaluated at 2000 randomly selected posterior samples. As the relative operator approximation error $\|\mathbf{A} - \widetilde{\mathbf{A}}\|/\|\mathbf{A}\|$ increases, the log-determinant values exhibit

*Table 4. Dimensions of parameters in Tests VI and VII in Section 4.2.*

| $d_z$ | $d_x$ | $d_u$ | $\widetilde{d}_u$ | $d_y$ |
|-------|-------|-------|-------------------|-------|
| 128 | $784\ (= 28^2)$ | $10^4\ (= 100^2)$ | $\{35^2,\ 20^2\}$ | 100 |

*Figure 12. Matrix constructions of $\mathbf{A}$ and $\widetilde{\mathbf{A}}$ in Tests VI and VII in Section 4.2. Left: $\mathbf{A} = \mathbf{O}\,\mathbf{F}\,\mathbf{P}$. Right: $\widetilde{\mathbf{A}} = \mathbf{O}\,\mathbf{P}_u\,\widetilde{\mathbf{F}}\,\mathbf{P}_x$. In our setting $d_y < d_x < d_u$, and $\widetilde{d}_u < d_u$.*

a systematic shift and increased spread, indicating stronger local volume distortion induced by the correction map. The right plots assess relative variability by considering ratios of Jacobian determinants between 2000 randomly selected pairs of approximate posterior samples from $\pi_a(x \mid y)$. Across all tests, these ratios remain tightly concentrated around one, with the 5%–95% quantile range remaining narrow even at larger approximation errors. This suggests that, despite changes in absolute scale, the Gauss–Newton transformation induces relatively uniform local volume changes across the posterior in the test.

### C.4. Nonlinear Helmholtz Test in Section 4.3

**Forward operator $\mathbf{A}_i(x)$ and its approximation $\widetilde{\mathbf{A}}_i(x)$.** The nonlinear forward model and its adjoint are introduced in Section B.2. The nonlinear Helmholtz operator is defined as

$$\mathcal{L}(x) := -\Delta - k^2 x.$$

The medium parameter $x$ is represented on a $32 \times 32$ uniform grid over the domain $[0,1] \times [0,1]$. The Helmholtz operator $\mathcal{L}(\cdot)$ is discretized on a finer $128 \times 128$ uniform grid using a second-order finite-difference scheme with homogeneous Neumann boundary conditions; we denote the resulting nonlinear discretized operator by $\mathbf{L}(\cdot)$. The wavenumber is set to $k = 3.4\pi$ in this test.

We consider $n_s = 4$ distinct source terms $\{f_i\}_{i \in [n_s]}$. For each source, the corresponding observation is generated by

$$y_i = \mathbf{O}\,\mathbf{L}^{-1}(\mathbf{P}x)\,f_i + e_i, \qquad i \in [n_s], \tag{81}$$

where $\mathbf{P}$ denotes a prolongation operator mapping the medium parameter from the coarse grid to the fine grid, and $\mathbf{O}$ is the observation operator. Accordingly, the nonlinear forward operator is defined as

$$\mathbf{A}_i(x) := \mathbf{O}\,\mathbf{L}^{-1}(\mathbf{P}x)\,f_i, \qquad i \in [n_s]. \tag{82}$$

To construct an approximate forward model, we introduce a coarser $64 \times 64$ grid for the wavefield discretization. Let $\widetilde{\mathbf{L}}(\cdot)$ denote the discretized nonlinear Helmholtz operator on this coarse grid. Analogous to the linear construction in Equation (80), we define the approximate nonlinear forward operator as

$$\widetilde{\mathbf{A}}_i(x) := \mathbf{O}\,\mathbf{P}_u\,\widetilde{\mathbf{L}}^{-1}(\mathbf{P}_x x)\,f_i, \qquad i \in [n_s], \tag{83}$$

where $\mathbf{P}_x$ and $\mathbf{P}_u$ are sparse prolongation operators used to map the medium parameter and wavefield, respectively, from coarse grids to the fine grid. Table 5 summarizes the dimensions of parameters involved in this test.

**GMRES solver with Neumann–DCT preconditioning.** We solve the (nonlinear) Helmholtz linear systems arising in Equation (82) and Equation (83) and related adjoint computations using GMRES with a matrix-free spectral preconditioner. The preconditioner is based on a constant-coefficient approximation of the Helmholtz operator,

$$\mathcal{M} := -\Delta + k^2\big(1 + \bar{x}\big),$$

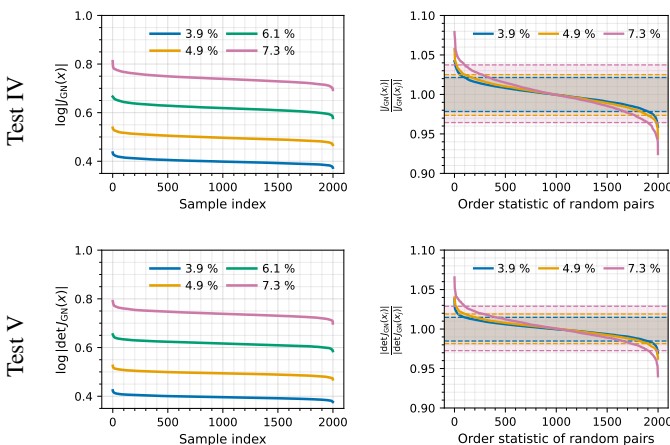

*Figure 13. Jacobian log-determinant diagnostics for the Gauss–Newton correction. Each row corresponds to one test case. **Left:** Sorted values of the log determinant of the Jacobian of a single Gauss–Newton transformation, evaluated at 2000 randomly selected posterior samples from $\pi_a(x \mid y)$. **Right:** Order statistics of ratios of Jacobian determinants between 2000 randomly selected pairs of posterior samples from $\pi_a(x \mid y)$, i.e., $\det J_{\mathrm{GN}}(x_i)/\det J_{\mathrm{GN}}(x_j)$. The numbers shown in the legends indicate the relative operator approximation error $\|\mathbf{A} - \widetilde{\mathbf{A}}\|/\|\mathbf{A}\|$. In the right panels, dashed lines denote the 5%–95% quantile range.*

*Table 5. Dimensions of parameters in the nonlinear Helmholtz test in Section 4.3.*

| $d_x$ | $d_u$ | $\widetilde{d}_u$ | $d_y$ |
|---|---|---|---|
| 1024 $(= 32^2)$ | 16,384 $(= 128^2)$ | 4,096 $(= 64^2)$ | 512 |

where $\bar{x}$ denotes the spatial average of the medium parameter $x$ and homogeneous Neumann boundary conditions are imposed. On a uniform $n \times n$ grid, the Neumann Laplacian is diagonalized by the two-dimensional discrete cosine transform (DCT). Let $\Lambda$ denote the eigenvalues of $-\Delta$ under Neumann boundary conditions. Applying $\mathcal{M}^{-1}$ to a vector $v$ is implemented by reshaping $v$ to an $n \times n$ array, computing its 2D DCT, dividing entrywise by $\Lambda + k^2(1 + \bar{x})$, and transforming back via the inverse 2D DCT. This results in an efficient $\mathcal{O}(n^2 \log n)$ preconditioner application that substantially accelerates GMRES convergence.

**Proposal for `Proximal-IMH`.** The objective of the proximal correction in this multiple-source setting is the following: let $\widetilde{x} \sim \pi_a$, the proposal of `Proximal-IMH` is

$$x \leftarrow \arg\min_{x} \sum_{i \in [n_s]} \left\| \mathbf{A}_i(x) - \widetilde{\mathbf{A}}_i(\widetilde{x}) \right\|^2 + \beta \|x - \widetilde{x}\|^2. \tag{84}$$

**One Gauss–Newton step formulation for Equation (84).** A single Gauss–Newton step to optimize Equation (84) initialized at $\widetilde{x}$ is given by

$$\begin{cases} x \leftarrow \widetilde{x} + \delta x, \\ \left(\sum_{i=1}^{n_s} \mathbf{J}_i(\widetilde{x})^\top \mathbf{J}_i(\widetilde{x}) + \beta \mathbf{I}\right) \quad \delta x = -\sum_{i=1}^{n_s} \mathbf{J}_i(\widetilde{x})^\top \left( \mathbf{A}_i(\widetilde{x}) - \widetilde{\mathbf{A}}_i(\widetilde{x}) \right), \end{cases} \tag{85}$$

where $\mathbf{J}_i(\widetilde{x}) := \left. \dfrac{\partial \mathbf{A}_i(x)}{\partial x} \right|_{x=\widetilde{x}}$ denotes the Jacobian of $\mathbf{A}_i$ evaluated at $\widetilde{x}$.

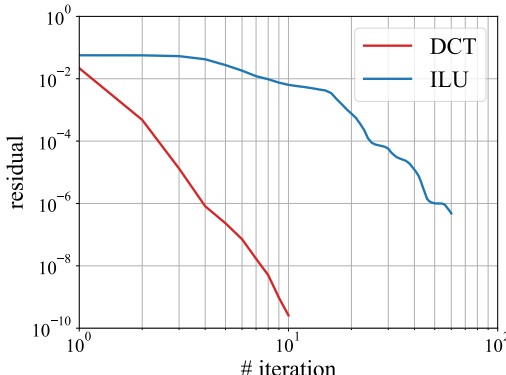

*Figure 14. Effect of preconditioning on GMRES convergence for the nonlinear Helmholtz test.*

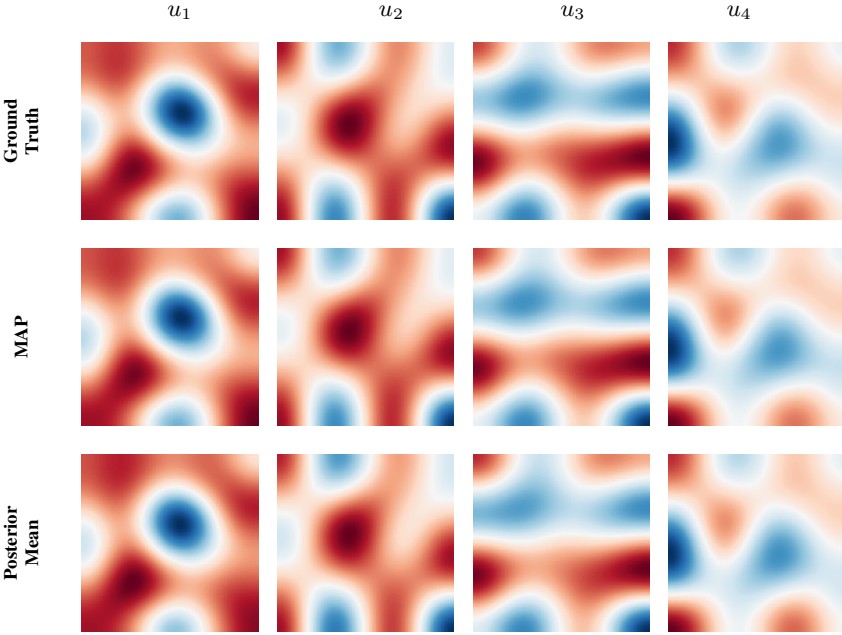

*Figure 15. Wavefields $\{u_i = \mathcal{L}(x)^{-1} f_i\}_{i=1}^4$ for the nonlinear Helmholtz test. Each column corresponds to a distinct source $f_i$. **Top:** Ground-truth wavefields. **Middle:** Wavefields obtained from the MAP estimate. **Bottom:** Posterior mean wavefields computed from posterior samples.*

