# OpenReview forum: "Proximal-IMH: Proximal Posterior Proposals for Independent Metropolis–Hastings with Approximate Operators"
_ICML.cc/2026/Conference — ICML 2026 regular_

### Official Review · Reviewer_Ue7A · 2026-03-09

**Soundness:** 3
**Presentation:** 3
**Significance:** 2
**Originality:** 2
**Overall Recommendation:** 4
**Confidence:** 3

**Summary:**

The paper addresses a limitation of the Latent-Independence Metropolis-Hastings (IMH) algorithm: the ill-conditioning that can arise from transforming the forward map into a square. Latent-IMH samples from the posterior of Bayesian inverse problems while exploiting a cheap approximate forward operator to reduce computational cost. The proposed alternative, Proximal-IMH, generates proposals by solving an auxiliary optimization problem that adjusts a sample from the approximate posterior so as to better match the exact data-misfit term, while penalizing large deviations from the original approximate sample. The method is studied in a simplified linear diagonal setting and then applied to nonlinear inverse problems via local Gauss-Newton transformation and additional heuristic approximations. Numerical experiments on both linear and nonlinear inverse problems support the improvements demonstrated by the theory.

**Compliance With Llm Reviewing Policy:**

Affirmed.

**Final Justification:**

I confirm my **weak accept** score. The authors clarified some missing details regarding the theoretical section. However, the fact that the nonlinear part remains largely heuristic and relies on approximate arguments, justified either empirically or only under restrictive assumptions, is a limitation of the work (see Reviewer An2Y on this point) and prevents me from giving it a higher score.

**Key Questions For Authors:**

- Is $K$ invertible in Test II?
- Are there nonlinear inverse problems in which a first-order local linearization, as used in the Gauss-Newton correction step, is insufficient, so that higher-order information or multiple correction steps would be needed for the proposal to remain accurate?

**Limitations:**

Yes.

**Strengths And Weaknesses:**

**Strengths:**

The paper---which I find well-written---proposes an improvement over an existing IMH method for Bayesian inverse problems. It clearly presents the motivations, i.e., a key limitation of Latent-IMH, and addresses it through a method supported by theoretical results in an idealized setting and by a series of numerical experiments on both linear and nonlinear inverse problems.

The results in the idealized setting are informative and help explain why the proposed method should improve over prior IMH variants. Finally, the paper does a good job in supporting these claims with numerical experiments, with applications to both linear and nonlinear inverse problems.

**Weaknesses:**

The paper is honest about its main limitation, i.e., non-linear inverse problems, where the analysis becomes mostly heuristic and relies on approximate arguments that are justified either empirically or under restrictive assumptions. I think the authors do their best to motivate the success of their method in nonlinear settings, even though the paper would have been much stronger with a more rigorous theoretical justification.

One weakness of the theoretical presentation is that a few assumptions seem to be left implicit. For example, in the linear setting, the paper uses often $K^{-1}$ explicitly, but it does not clearly state conditions ensuring that $K$ is invertible.

---

> ### Author Rebuttal · Authors · 2026-03-31
>
> We thank Reviewer Ue7A for the detailed reading and very constructive comments. We clarify them as follows:
>
> ---
> -- **K invertibility:**
>
> We thank the reviewer for pointing this out. We agree that the invertibility condition for $K$ should be stated explicitly in the linear theory section. In Eq. (4), $\mathbf{K} = (\mathbf{A}^\top \mathbf{A} + \beta \mathbf{I})^{-1}(\mathbf{A}^\top \widetilde{\mathbf{A}} + \beta \mathbf{I})$. Since $\beta > 0$, $\mathbf{A}^\top \mathbf{A} + \beta \mathbf{I}$ is always symmetric positive definite. Thus, $\mathbf{K}$ is invertible if and only if $\mathbf{A}^\top \widetilde{\mathbf{A}} + \beta \mathbf{I}$ is invertible, equivalently if and only if $-\beta \notin \sigma(\mathbf{A}^\top \widetilde{\mathbf{A}})$. We will add this condition explicitly in the revision.
>
> We also note that this assumption is mild in the regime of interest. When $\widetilde{\mathbf{A}} = \mathbf{A}$, we have $\mathbf{K} = \mathbf{I}$. More generally, if $\widetilde{\mathbf{A}}$ is a sufficiently small perturbation of $\mathbf{A}$, then $\mathbf{A}^\top \widetilde{\mathbf{A}} + \beta \mathbf{I}$ remains invertible by continuity of the spectrum. This is precisely the setting studied in our analysis, where the approximation quality is controlled by $\Delta \mathbf{A} = \widetilde{\mathbf{A}} - \mathbf{A}$ and the bounds depend on moderate operator mismatch. In practice, $\beta$ is a tunable regularization parameter already chosen relative to the noise level, and one can choose it away from pathological values where $-\beta$ coincides with an eigenvalue of $\mathbf{A}^\top \widetilde{\mathbf{A}}$. Equivalently, for fixed $\mathbf{A}$ and $\widetilde{\mathbf{A}}$, non-invertibility can occur only for the exceptional values $\beta = -\lambda$ with $\lambda \in \sigma(\mathbf{A}^\top \widetilde{\mathbf{A}})$, so avoiding such values is not restrictive in practice.
>
>
> ---
> -- **K invertible in Test II?**
>
> Yes. In the original submission, we reported both $\lVert \mathbf{I} - \widetilde{\mathbf{F}}^{-1}\mathbf{F} \rVert$ and $\lVert \mathbf{I} - \mathbf{K}^{-1} \rVert$ for Tests I--III in Figure 3. To make invertibility explicit, we additionally report below in Table 1 the minimum singular value and condition number of $\mathbf{K}$ in Test II for several operator perturbation levels. In all reported cases, $\sigma_{\min}(\mathbf{K})$ remains strictly positive, confirming that $\mathbf{K}$ is invertible in these Test II instances.
>
> **Table 1:** Minimum singular value and condition number of $\mathbf{K}$ in Test II.
>
> | $\frac{\lVert \mathbf{A} - \widetilde{\mathbf{A}}\rVert}{\lVert \mathbf{A}\rVert}$ (%) | 0.7 | 1.4 | 2.7 | 4.1 | 6.8 | 13.5 |
> |:---:|:---:|:---:|:---:|:---:|:---:|:---:|
> | $\sigma_{\min}(\mathbf{K})$ | 0.92 | 0.86 | 0.73 | 0.64 | 0.49 | 0.29 |
> | $\kappa(\mathbf{K})$ | 1.1 | 1.4 | 1.9 | 2.6 | 4.4 | 12.6 |
>
> ---
>
> ---
>
> -- **Gauss--Newton (GN) steps:**
>
> This is a very good practical question. In sufficiently nonlinear inverse problems, a single GN correction may be insufficient, while additional GN steps can improve proposal quality at increased cost. Thus, the appropriate number of correction steps depends on the problem, including the quality of the approximate operator, the available computational budget, and the marginal gain from extra correction steps.
>
> To study this, we ran additional experiments for Test VI with one, two, and three GN steps (see Table 2 in below). The table shows that when the approximation error is larger, moving from one step to two steps significantly improves both acceptance and relative mean error, while a third step provides only marginal additional benefit. When the approximation error is smaller (bottom table), one step already performs well and additional steps yield only modest gains. This shows a clear cost--accuracy tradeoff.
>
> In practice, the number of GN steps can be selected via a short pilot run by monitoring *acceptance rate* and the GN objective (Eq. (3b)). We will add this discussion and the table to clarify that one GN step is a practical default rather than a universally optimal choice.
>
>
> **Table 2:** Effect of GN steps in Proximal-IMH for Test VI. Relative mean error after 60K MH steps. Top: $\frac{\lVert \mathbf{A} - \widetilde{\mathbf{A}}\rVert}{\lVert \mathbf{A}\rVert}$ = 4.6\%.Bottom: 3.4\%.
>
> | # GN Steps | Loss | Acceptance Ratio (\%) | Relative Mean Error (\%) |
> |:---:|:---:|:---:|:---:|
> | 1 | 4.6E-4 | 46.2 $\pm$ 0.5 | 3.5 $\pm$ 0.1 |
> | 2 | 4.0E-4 | 58.4 $\pm$ 0.3 | 2.5 $\pm$ 0.05 |
> | 3 | 4.0E-4 | 59.3 $\pm$ 0.1 | 2.4 $\pm$ 0.04 |
>
> | # GN Steps | Loss | Acceptance Ratio (\%) | Relative Mean Error (\%) |
> |:---:|:---:|:---:|:---:|
> | 1 | 1.5E-4 | 63.2 $\pm$ 0.2 | 1.8 $\pm$ 0.04 |
> | 2 | 1.1E-4 | 71.9 $\pm$ 0.1 | 1.6 $\pm$ 0.04 |
> | 3 | 1.1E-4 | 72.5 $\pm$ 0.1 | 1.6 $\pm$ 0.02 |
>
> ---
> We thank the reviewer again for these helpful comments; in the revision, we will state the invertibility condition explicitly and add the above clarifications and tables.

---

> > ### Author Rebuttal · Reviewer_Ue7A · 2026-04-03
> >
> > I thank the authors for their answer. My concerns have been resolved. I trust the authors will state the invertibility conditions explicitly in the main text.

---

> > > ### Author Response · Authors · 2026-04-07
> > >
> > > Thank you for the positive feedback. We will add these clarifications into the revised manuscript.

---

### Official Review · Reviewer_AKS6 · 2026-03-09

**Soundness:** 3
**Presentation:** 4
**Significance:** 4
**Originality:** 3
**Overall Recommendation:** 5
**Confidence:** 3

**Summary:**

The paper proposes a significant improvement to using approximate posteriors to guide proposals in Independent Metropolis-Hastings. The core idea is to use approximate and quick posterior to guide the proposal, and then apply a simple optimisation step within the proposal to correct for approximate posterior mismatch. The paper demonstrates impressive improvement of performance on several linear and non-linear benchmarks.

**Compliance With Llm Reviewing Policy:**

Affirmed.

**Key Questions For Authors:**

No questions really, it is an excellent paper.
But if adding 1000 step performance comparison is not too difficult, it will be greatly appreciated.

**Limitations:**

Yes

**Strengths And Weaknesses:**

Strengths: The paper proposes a significant improvement in methodology for IMH sampling. It is clearly written and easy to understand. The results are very impressive.

Weaknesses: To enable inference over non-linear models, authors had to discard the Jacobean ratio justified by its insignificance. Authors emphasise the need to monitor the scale of the Jacobean ratio for particular applications to make sure such simplification is appropriate.

It would have been nice to see a comparison table to see computational time required to make 1000 steps using Latent-IMH, Proximal-IMH and NUTS. Particular interest is whether optimisation in (3b) is more expensive than latent variable sampling in Latent-IMH. There are clear theoretical advantages for using Proximal approach even if it is slower.

---

> ### Author Rebuttal · Authors · 2026-03-31
>
> We thank Reviewer AKS6 for the positive and insightful review, and for recognizing the methodological contribution, clarity, and strong empirical performance of our approach. We also appreciate the suggestion regarding cost/sample comparison.
>
> 1.  We summarize the main properties and cost/sample of each method in Table 1:
>
> - **Approx-IMH:** Given $\widetilde{x} \sim \pi_a(x\mid y)$, only one forward evaluation $A\widetilde{x}$ is required to compute the acceptance ratio.
>  - **Latent-IMH:** The proposal requires solving $F x = \widetilde{F} \widetilde{x}$ using an iterative solver (CG if $F$ is SPD, otherwise GMRES). With preconditioning (standard in PDE problems), the cost is $n_{\text{iter}} C_f$, where $n_{\text{iter}}$ depends on conditioning and the preconditioner.
>
> - **Proximal-IMH:** The proximal correction solves $(A^\top A + \beta I)x = (A^\top \widetilde{A} + \beta I)\widetilde{x}$, an **SPD** system. Each CG iteration requires one forward and one adjoint solve. The system can be efficiently preconditioned (e.g., using $\widetilde{A}^\top \widetilde{A} + \beta I$), consistent with standard PDE-constrained optimization techniques (e.g., multigrid). In practice, this leads to a small number of iterations, so the cost is approximately $2 n_{\text{iter}} C_f$.
>
> - **NUTS:** Cost depends on the tree depth $n_\ell$. Each leapfrog step requires one gradient evaluation (one forward + one adjoint solve), and the number of steps scales as $2^{n_\ell}$. In our experiments, we use $n_\ell \approx 7\text{--}9$, which is required to achieve stable sampling with acceptance rates around 80–85%.
>
> **Table 1:** Comparison between methods. $C_f$ is the cost of one forward apply; we assume that the adjoint has the same cost. $n_{\text{iter}}$ and $n_{\text{CG}}$ denote the number of solver iterations; $n_\ell$ denotes the NUTS tree depth.
>
> | | Approx-IMH | Latent-IMH | Proximal-IMH | NUTS|
> |:---:|:---:|:---:|:---:|:---:|
> |nonsquare F | $\checkmark$ | $\times$  |$\checkmark$  |$\checkmark$ |
> | nonlinear F| $\checkmark$  | $\times$  | $\checkmark$  |$\checkmark$  |
> | independent proposal| $\checkmark$   |$\checkmark$   |$\checkmark$   |$\times$  |
> |cost/sample | $C_f$ | $n_{\text{iter}} C_f$  |  $2n_{\text{CG}} C_f$| $2^{n_{\ell}+1} C_f$|
>
> 2. Cost/proposal comparison between Latent-IMH and Proximal-IMH:
>
> Tables 2–3 report the number of CG/PCG iterations per proposal in Test I and Test II. For Latent-IMH, we use $\widetilde{F}$ as a preconditioner; for Proximal-IMH, we use $\widetilde{A}^\top \widetilde{A} + \beta I$.
>
> In Test I, both methods benefit from preconditioning: iteration counts are reduced to small constants (2–6). Latent-IMH can be slightly cheaper in some cases, but the difference is modest.
>
> In Test II, with PCG, Proximal-IMH maintains consistently low iteration counts (2–7), while Latent-IMH becomes significantly more expensive (30–99 iterations). This is because $\widetilde{F}^{-1} F$ becomes poorly conditioned under the perturbation in Test II, whereas $\widetilde{A}^\top \widetilde{A} + \beta I$ an effective preconditioner for the proximal solve.
>
> Overall, the relative cost between Latent-IMH and Proximal-IMH per proposal is problem-dependent and largely determined by the effectiveness of preconditioning.
>
> Besides, as also noted by the reviewer, Proximal-IMH provides clear theoretical advantages.
>
> **Table 2 :** Number of CG or preconditioned CG (PCG) iterations in **Test I** for Latent-IMH and Proximal-IMH per proposal. Relative residual tolerance is set to 1.e-5.
>
> |$\frac{\lVert A- \widetilde{A}\rVert}{\lVert A \rVert}$ | 5.8% | 9.2% | 16.2%|23.1%|
> |:---:|:---:|:---:|:---:|:---:|
> |Latent: CG #iter |16 |17 |18 | 19|
> |Latent: PCG #iter |2 | 2|3 | 3|
> |Proximal: CG #iter |8 | 8| 12| 12 |
> |Proximal: PCG #iter |3 | 3| 4| 6|
>
>
> **Table 3 :** Number of CG or preconditioned CG (PCG) iterations in **Test II** for Latent-IMH and Proximal-IMH per proposal. Relative residual tolerance is set to 1.e-5.
>
> |$\frac{\lVert A- \widetilde{A}\rVert}{\lVert A \rVert}$ |1.2% | 2.7% | 8.3%|16.5%|
> |:---:|:---:|:---:|:---:|:---:|
> |Latent: CG #iter |18 |19 |24|25|
> |Latent: PCG #iter |30 | 34|58 | 99|
> |Proximal: CG #iter |9 | 12| 15| 17 |
> |Proximal: PCG #iter |2 | 3| 7| 7|
>
> 3. Regarding the suggestion to report wall-clock time for 1000 steps, we agree this can be informative. However, in PDE-constrained inverse problems, runtime is highly dependent on implementation details (e.g., solver tolerance, preconditioning, and hardware), making such comparisons less transferable.
> Instead, we report cost in units of forward/adjoint solves and iteration counts, which are standard and hardware-independent metrics. These results highlight that performance is primarily governed by conditioning and preconditioning, and can vary significantly across problems.
>
> ---
> We will include this discussion in the revised manuscript to clarify how cost should be interpreted in practice.

---

> > ### Author Rebuttal · Reviewer_AKS6 · 2026-04-05
> >
> > Thank you, my concerns are addressed, please consider adding these explanations to the manuscript.

---

> > > ### Author Response · Authors · 2026-04-07
> > >
> > > Thank you for the positive feedback. We will add these clarifications into the revised manuscript.

---

### Official Review · Reviewer_Cd7J · 2026-03-10

**Soundness:** 4
**Presentation:** 3
**Significance:** 3
**Originality:** 3
**Overall Recommendation:** 5
**Confidence:** 4

**Summary:**

This paper a new independence-MH algorithm, proximal-IMH, designed for bayesian inverse problems where the exact forward operator is computationally expensive. Proximal-IMH can be viewed as the approximate-IMH algorithm (proposal from a cheap, biased approximate posterior sample) with addtional correction steps through a proximal optimization step.

- For linear inverse models, the authors prove that Proximal-IMH achieves a lower expected KL divergence than standard approximate proposals and faster mixing times than Latent-IMH.

- For non-linear inverse model, the author extends the algorithm using a single Gauss-Newton step to approximate the proximal correction.

- The practical advantage of the proposed methods is justified through extensively experiments.

**Compliance With Llm Reviewing Policy:**

Affirmed.

**Key Questions For Authors:**

- The theoretical analysis shows that the Latent-IMH mixing time bound is independent of the observation noise variance $\sigma^2$. How should I understand this?

- How do you position this proximal-IMH relative to the established Proximal sampler [1] in the log-concave sampling literature?


Reference:
[1]: Lee, Y.T., Shen, R. &amp; Tian, K.. (2021). Structured Logconcave Sampling with a Restricted Gaussian Oracle. COLT.

**Limitations:**

yes

**Strengths And Weaknesses:**

## Strength

In my perspective, this work is solid and sound. The proposed methodology is examined through careful theoretical anlaysis (at least in the linear case) and empirical evaluations.  Here are some of the key strength of this work:

-  Rigorous theory for linear operators: For linear forward models with Gaussian noise and priors, the authors derive exact, closed-form expressions for the expected KL divergence (Theorem 3.2) and explicit bounds on the mixing time (Theorem 3.3). Both results intuitively explain why proximal-IMH is more robust to operator perturbations and converges faster than existing methods. Though the assumption of Thm 3.2 might be unrealistic, it does serve the purpose for demonstrating the idea.

- Resolving the key limitation of latent-IMH: previous work---latent-IMH---requires the forward operator to be transformed into a square, invertible matrix, as well as constructing and sampling from an approximate distribution in a high-dimensional latent space. The proposed proximal-IMH bypass these constraints by cleverly formulating the correction directly in the parameter space using an auxiliary optimization problem.

- Strong empirical performance.

## Weakness

- Choice of $\beta$: the performance of proximal-IMH relies on a proper tuning of $\beta$. While through theory, it seems recommended to use $\beta = \sigma^2$, Fig. 4 shows that performance degrades significantly if $\beta$ is even slightly over-regularized (> 2, 3 $\sigma$^2).  In practice, is there a principled way to tune $\beta$?

-  In high-dimensional cases, computing the correction opeartor $K$ involves inverting $(A^T A + \beta I)$. In my opinion, this can be quite expensive and may even outweigh the cost of evaluating the exact linear operator $F$. I wonder how would the author justify the cost of this inverse.

- Loss of exactness in non-linear extension. In the non-linear setting, the authors apply a Gauss-Newton map to the approximate sample to generate a proposal. However, in the practical implementation, the author explicitly neglect the jacobian determinant, which resulting in a biased sampler. Even though the author justified this by arguing that the determinant ratio is likely ~1, it can be seen from Fig 6 (test VI and VII) that the proximal-IMH struggle to achieve high accuracy comapred to NUTS.

- Insufficiency of a single Gauss-Newton step in non-convex landscapes. The gauss-newton methods achieve local quadratic convergence only when initialized sufficiently close to the optimum and when the residual is small. In constrast,  in highly non-linear or non-convex inverse problems, single linear approximation step is often insufficient to pull a sample from a biased approximate posterior into the high-probability region of the exact posterior.  As admitted by the author in the MNIST examples, "We observe that in several cases the reconstructed MAP point is wrong. This is due to the nonconvexity of solving for a MAP point".

---

> ### Author Rebuttal · Authors · 2026-03-31
>
> We thank Reviewer Cd7J for the detailed reading and very constructive comments. We clarify them as follows:
>
> ---
> -- **Choice of $\beta$:** We clarify that the paper recommends $\beta=\Theta(\sigma^2)$, not a fixed $\beta=\sigma^2$ (lines 139–150). In experiments, we use $\beta=\sigma^2$ as a default choice, without tuning, to demonstrate robustness without careful parameter selection.
> - **Practical tuning:** *Figures 4 and 7* show that the optimal $\beta$ for *acceptance rate* aligns with that for *relative mean error*. Thus, $\beta$ can be tuned via acceptance rate using a short pilot run (few hundred IMH steps).
> ---
> -- **Cost of computing $K$:**   We do not form or invert $K$ explicitly. Instead, applying $x = K^{-1}\widetilde{x}$ amounts to solving   $(A^\top A + \beta I)x = (A^\top \widetilde{A} + \beta I)\widetilde{x}$, which is an SPD system. We solve it with CG or PCG, where each iteration requires one forward and one adjoint solve. The iteration number depends on the conditioning of $A^\top A + \beta I$ and the preconditioner. For example we can use $P = \widetilde{A}^\top \widetilde{A} + \beta I$, whose application only uses the cheaper approximate operator. In PDE-based settings, this follows standard preconditioning ideas from PDE-constrained optimization [1] and in practice leads to only a small number of iterations. Moreover, because the proposal is independent, these computations can be *precomputed and parallelized*, which further improves scalability.
>
> [1] Rees, T., Dollar, H. S., and Wathen, A. J. Optimal solvers for PDE-constrained optimization. SIAM Journal on Scientific Computing, 32(1):271–298, 2010.
>
> ---
> -- **Non-linear extension:** We agree that neglecting the Jacobian determinant introduces bias. We explicitly note this limitation and recommend *monitoring the determinant ratio* (lines 228–230). This is problem-dependent and is explicitly acknowledged in conclusion section.
> - **Diagnostic:** In *Figure 11*, the determinant ratio is close to one for nonlinear Tests IV and V. In practice, this can be checked inexpensively using a small pilot run (e.g., a few hundred points); if it deviates, it should be included in the acceptance ratio.
> ---
> -- **Gauss–Newton (GN) steps:** A single GN step may be insufficient in highly nonlinear settingst. Additional steps can improve accuracy but increase cost, so the optimal number depends on *the quality of the approximate operator, computational budget, and marginal benefit*.
> - **Evidence and practice:** For Test VI (table below), two steps significantly improve acceptance and error over one step when approximation error is large , while a third step gives marginal gains. For smaller error, one step already performs well. In practice, similar to tuning $\beta$, the number of GN steps can be selected via a short pilot run by monitoring acceptance rate and the GN objective (Eq. (3b)).
>
>
>
> **Table:** Effect of GN steps in Proximal-IMH for Test VI. Relative mean error after 60K MH steps. Top: $\frac{\lVert \mathbf{A} - \widetilde{\mathbf{A}}\rVert}{\lVert \mathbf{A}\rVert}$ = 4.6\%.Bottom: 3.4\%.
>
> | # GN Steps | Loss | Acceptance Ratio (\%) | Relative Mean Error (\%) |
> |:---:|:---:|:---:|:---:|
> | 1 | 4.6E-4 | 46.2 $\pm$ 0.5 | 3.5 $\pm$ 0.1 |
> | 2 | 4.0E-4 | 58.4 $\pm$ 0.3 | 2.5 $\pm$ 0.05 |
> | 3 | 4.0E-4 | 59.3 $\pm$ 0.1 | 2.4 $\pm$ 0.04 |
>
> | # GN Steps | Loss | Acceptance Ratio (\%) | Relative Mean Error (\%) |
> |:---:|:---:|:---:|:---:|
> | 1 | 1.5E-4 | 63.2 $\pm$ 0.2 | 1.8 $\pm$ 0.04 |
> | 2 | 1.1E-4 | 71.9 $\pm$ 0.1 | 1.6 $\pm$ 0.04 |
> | 3 | 1.1E-4 | 72.5 $\pm$ 0.1 | 1.6 $\pm$ 0.02 |
>
> ---
>
> -- **Latent-IMH mixing time:** By Eqs. (2a) and (2c), the likelihood terms cancel in the MH acceptance ratio for Latent-IMH. Hence, the acceptance ratio depends only on the prior and the operator mismatch, and is independent of the noise variance $\sigma^2$.
>
> The same holds theoretically for the mixing-time bound, which is controlled by the local Lipschitz constant of the log weight
>
> $\log \frac{\pi_l(x|y)}{\pi(x|y)} = \log p(x) - \log p(\widetilde{F}^{-1} F x)$,
>
> which does not involve the likelihood. Therefore, its local Lipschitz constant, and thus the mixing-time bound, are independent of $\sigma^2$.
>
> ---
> --**Proximal sampler:** Lee et al. (2021) studies proximal samplers for log-concave distributions, where the proximal operator is applied to the negative log-density. In contrast, Proximal-IMH targets Bayesian inverse problems with approximate forward operators. Our proximal step is a data-consistency correction (Eq. (3b)), not a proximal update of the log-density. Algorithmically, Proximal-IMH is a **global independence MH** method, while proximal samplers are **local MCMC chains**. Thus, the two approaches address different settings and are complementary.
>
> ---
> We thank the reviewer again for these helpful comments. We will incorporate these clarifications, diagnostics, and the additional GN-step results in the revised version.

---

> > ### Author Rebuttal · Reviewer_Cd7J · 2026-04-03
> >
> > The authors' response has addressed most of my concerns. Though, my concern regarding the inexactness of the sampler in the non-linear case is not fully addressed. I share the same thoughts as Reviewer An2Y, where the justification of Jacobian close to 1 is empirical, and lack proper error quantification.

---

> > > ### Author Response · Authors · 2026-04-07
> > >
> > > We thank the reviewer for the additional comments. We agree that, in the nonlinear setting, omitting the Jacobian determinant ratio is a practical approximation and may introduce bias; our current justification for neglecting this term is empirical and does not provide a general error quantification. By including the determinant ratio, exactness can be recovered, but at additional computational cost. In large-scale settings, this cost may be reduced using matrix-free randomized algorithms for computing the log-determinant [1,2]. We will add this discussion in the revised manuscript.
> > >
> > > [1] Saibaba et al. “Randomized matrix-free trace and log-determinant estimators.” _Numerische Mathematik_, 2017.
> > > [2] Knoll and Keyes. “Jacobian-free Newton–Krylov methods: A survey of approaches and applications.” _JCP_, 2004.

---

### Official Review · Reviewer_An2Y · 2026-03-12

**Soundness:** 2
**Presentation:** 3
**Significance:** 2
**Originality:** 3
**Overall Recommendation:** 3
**Confidence:** 5

**Summary:**

The paper studies sampling for Bayesian inverse problems where evaluating the exact forward operator is expensive but a cheaper approximation is available. The authors propose Proximal-IMH, an independence Metropolis--Hastings sampler that generates proposals from an approximate posterior and then applies a proximal correction to reduce the mismatch between the approximate and exact models. In the linear case the correction has a closed-form expression based on a regularized least-squares problem. For nonlinear operators the method applies a single Gauss–Newton update to obtain a corrected proposal. The paper provides theoretical analysis for linear forward models, including bounds on the KL divergence between proposal and target distributions and mixing-time guarantees. Experiments on multimodal synthetic targets, imaging inverse problems with MNIST-based priors, and nonlinear Helmholtz PDE inverse problems show improved acceptance rates and faster convergence compared with several IMH baselines.

**Compliance With Llm Reviewing Policy:**

Affirmed.

**Final Justification:**

The limitations are still profound. I will not change my initial score.

**Key Questions For Authors:**

1. How does the cost of solving the proximal correction compare with the cost of evaluating the forward operator in large PDE-based inverse problems? Clarifying this would help assess the scalability of the method.

2. The nonlinear extension ignores Jacobian determinant terms in practice. How does this approximation affect the correctness of the resulting Markov chain?

3. Can the method be compared with more recent surrogate-based or learned proposal approaches used in large-scale Bayesian inference? Such comparisons would help position the method relative to current alternatives.

**Limitations:**

Yes

**Strengths And Weaknesses:**

I found the problem well motivated. Many inverse problems rely on approximate forward models because exact evaluations are too expensive. The idea of correcting samples from an approximate posterior with a proximal step is simple and intuitive. It provides a clear way to combine approximate models with exact inference. I also liked the theoretical analysis for the linear case. The KL divergence bounds help explain why the proximal proposal can be closer to the true posterior than the approximate one. The mixing-time discussion also gives some insight into how operator errors affect the performance of IMH schemes. The experiments are reasonably broad. They include multimodal targets, imaging inverse problems with deep priors, and nonlinear PDE models. In many cases the method achieves higher acceptance rates and converges faster than standard IMH proposals.

Despite the strengths mentioned above, I have some concerns. Most of the theoretical guarantees rely on linear forward models with strong assumptions such as Gaussian noise and log-concave priors. Many inverse problems are highly nonlinear, and the nonlinear extension here uses a single Gauss--Newton step while ignoring Jacobian determinant terms in practice. Because of this, it is not clear how strong the guarantees remain in the nonlinear setting. Another point is computational cost. Each proposal requires solving an auxiliary optimization problem. The paper argues that this is cheap relative to evaluating the forward operator, but the cost may still grow in high dimensions or when multiple linear solves are required. I also wondered about comparisons with more recent proposal methods. The experiments focus mainly on IMH variants, while recent work on surrogate-based MCMC, normalizing-flow proposals, and diffusion-based samplers could provide additional context. Finally, some of the theoretical results are derived in simplified Gaussian settings, and it is not fully clear how these insights carry over to the more complex experiments.

---

> ### Author Rebuttal · Authors · 2026-03-31
>
> We thank Reviewer An2Y for the thoughtful comments and helpful questions.
>
> 1. **Cost of proximal vs. forward operator.**
>
> In our method, the proximal correction solves $(A^\top A + \beta I)x = (A^\top \widetilde{A} + \beta I)\widetilde{x}$, an SPD system. We solve it with CG or PCG, where each iteration requires one forward and one adjoint solve. The iteration number depends on the conditioning of $A^\top A + \beta I$ and the preconditioner. For example we can use $P = \widetilde{A}^\top \widetilde{A} + \beta I$, whose application only uses the cheaper approximate operator. In PDE-based settings, this follows standard preconditioning ideas from PDE-constrained optimization [1] and in practice leads to only a small number of iterations. Moreover, because the proposal is independent, these computations can be *precomputed and parallelized*, which further improves scalability.
>
> [1] Rees, T., Dollar, H. S., and Wathen, A. J. Optimal solvers for PDE-constrained optimization. SIAM Journal on Scientific Computing, 32(1):271–298, 2010.
>
> ---
> 2. **Nonlinear extension and Jacobian determinant.**
>
> We agree that neglecting the Jacobian determinant introduces bias, and we explicitly acknowledge this limitation in the paper. Our rationale is that when $A \approx \widetilde{A}$, the Gauss--Newton update is a small local correction around $\widetilde{x}$, so the determinant ratio is expected to be close to one. We also analyze a representative nonlinear model in Section 3.4 and recommend monitoring this ratio in practice.
>
> Empirically, *Figure 11* shows that for nonlinear Tests IV and V, the determinant ratio stays close to one for most samples, indicating that its effect is negligible in these settings.
>
> In practice, this can be checked inexpensively on a small pilot run; if the ratio is not negligible, it should be included in the acceptance probability to recover correctness.
>
> ---
>
> 3. **Gauss–Newton (GN) steps.**
>
> We agree that a single GN step may be insufficient in highly nonlinear settings. To study this, we ran additional experiments for Test VI with one, two, and three GN steps (see table bleow). The table shows that when the approximation error is larger, moving from one step to two steps significantly improves both acceptance and relative mean error, while a third step provides only marginal additional benefit. When the approximation error is smaller (bottom table), one step already performs well, and additional steps yield only modest gains. This highlights a clear *cost–accuracy tradeoff*.
>
> In practice, the number of GN steps can be selected via a short pilot run by monitoring *acceptance rate* and the GN objective (Eq. (3b)). We will add this discussion and the table to clarify that one GN step is a practical default rather than a universally optimal choice.
>
> **Table:** Effect of GN steps in Proximal-IMH for Test VI. Relative mean error after 60K MH steps. Top: $\frac{\lVert \mathbf{A} - \widetilde{\mathbf{A}}\rVert}{\lVert \mathbf{A}\rVert}$ = 4.6\%. Bottom: 3.4\%.
>
> | # GN Steps | Loss | Acceptance Ratio (\%) | Relative Mean Error (\%) |
> |:---:|:---:|:---:|:---:|
> | 1 | 4.6E-4 | 46.2 $\pm$ 0.5 | 3.5 $\pm$ 0.1 |
> | 2 | 4.0E-4 | 58.4 $\pm$ 0.3 | 2.5 $\pm$ 0.05 |
> | 3 | 4.0E-4 | 59.3 $\pm$ 0.1 | 2.4 $\pm$ 0.04 |
>
> | # GN Steps | Loss | Acceptance Ratio (\%) | Relative Mean Error (\%) |
> |:---:|:---:|:---:|:---:|
> | 1 | 1.5E-4 | 63.2 $\pm$ 0.2 | 1.8 $\pm$ 0.04 |
> | 2 | 1.1E-4 | 71.9 $\pm$ 0.1 | 1.6 $\pm$ 0.04 |
> | 3 | 1.1E-4 | 72.5 $\pm$ 0.1 | 1.6 $\pm$ 0.02 |
>
> ---
>
> 4. **Relation to surrogate-based and learned proposal methods.**
>
> We agree that surrogate-based MCMC, flow-based proposals, and diffusion-based samplers are important related directions. Our focus here is narrower: Proximal-IMH studies how to use a cheaper approximate operator to construct an effective independence proposal with exact MH correction, so our most direct comparisons are with IMH methods exploiting the same approximate-model structure.
>
> Among these directions, surrogate-based approaches are the most directly complementary to our framework, since a learned surrogate or neural operator can serve as the approximate forward map $\widetilde{F}$ (or $\widetilde{A}$). Likewise, if an approximate operator is available, a normalizing flow could be used to learn a fast sampler for the induced approximate posterior $\pi_a(x\mid y)$, after which Proximal-IMH can be used as a second-stage correction toward the exact posterior. By contrast, diffusion-based samplers are less directly aligned with our setting, since many do not provide a tractable proposal density for standard independence-MH correction.
>
> We agree that a broader comparison would be valuable, but carrying it out carefully, or developing such hybrid methods, would require substantial additional methodological and implementation work beyond the scope of the present paper.

---

> > ### Author Rebuttal · Reviewer_An2Y · 2026-04-03
> >
> > The rebuttal provides additional clarification, but my concerns are only partially addressed. On computational cost, while the authors argue that the proximal step can be solved efficiently using CG or PCG, each iteration still requires forward and adjoint solves. In large-scale PDE settings, this cost is nontrivial, and the claim that only a small number of iterations suffices remains largely heuristic without concrete runtime comparisons. The argument that proposals can be parallelized is standard for IMH and does not directly address the per-sample computational burden.
> >
> > Regarding the nonlinear extension, the acknowledgment that neglecting the Jacobian determinant introduces bias is appreciated, but this raises a fundamental issue: without this term, the Markov chain is not guaranteed to target the correct posterior. The justification that the determinant ratio is close to one is empirical and problem-dependent, and does not provide a general guarantee. Similarly, while additional experiments with multiple Gauss--Newton steps are helpful, they highlight a nontrivial cost of accuracy tradeoff and introduce additional tuning, which is not fully resolved.
> >
> > More broadly, some of my original concerns remain unaddressed. In particular, the theoretical results rely on linear forward models with Gaussian noise and strong convexity assumptions, and it is still unclear how these guarantees extend to the nonlinear and more complex settings considered in the experiments. The connection between the theoretical analysis and the empirical results therefore remains somewhat limited.
> >
> > Finally, while the authors clarify their focus on IMH methods with approximate operators, the lack of comparison with more recent surrogate-based or learned proposal approaches limits clarity on broader relevance of the method in modern large-scale Bayesian inference.

---

> > > ### Author Response · Authors · 2026-04-07
> > >
> > > We thank the reviewer for the additional comments. Our responses are below.
> > >
> > > **(1) Cost comparisons**
> > >
> > > We report in **Tables 1 and 2** the **number of exact forward solves**  required by each method to reach a target relative mean error. All results for Proximal-IMH use **unpreconditioned** CG, so the reported costs are conservative.  We also summarize the per-sample cost in _Table 1 of our rebuttal to Reviewer AKS6_.
> > >
> > > We thank the reviewer for raising this point and will include these comparisons in the revision. Across the tests in Tables 1 and 2, Proximal-IMH is consistently much *cheaper* than the competing methods in *total forward-solve cost*. Compared with NUTS, it lowers total cost by roughly 1–2 orders of magnitude while retaining an independence proposal. Compared with other IMH methods, it is also more robust both theoretically and empirically.
> > >
> > > **Table 1.** Number of exact forward solves required to reach **5%** relative mean error. For **Proximal-IMH**, CG is **unpreconditioned** with relative residual tolerance $10^{-3}$. Tests IV/V use operator error 3.9%, and Tests VI/VII use 3.4%. “---” means Approx-IMH did not reach **5%** error within 2.5E5 samples.
> > >
> > > | Test | Approx-IMH | Proximal-IMH | NUTS |
> > > |:---:|:---:|:---:|:---:|
> > > | Test IV | --- | 1.4E5 | 1.5E6 |
> > > | Test V | 1.7E5 | 8.4E4 | 1.2E6 |
> > > | Test VI | 1.8E5 | 3.8E4 | 1.4E6 |
> > > | Test VII | --- | 5.6E4 | 1.3E6 |
> > > | Nonlinear Helmholtz (Figure 8) | --- | 7.2E4 | 5.7E5 |
> > >
> > > **Table 2.** Number of exact forward solves required to achieve 10\% and 5\% relative mean error in Test I. “---” means the method did not reach the target error within 2.5E5 samples.
> > >
> > > | Test | Approx-IMH | Latent-IMH | Proximal-IMH | NUTS |
> > > |:---:|:---:|:---:|:---:|:---:|
> > > | Test I (mean error: 10\%) | --- | 2.3E6 | 1.5E4 | 6.7E5 |
> > > | Test I (mean error: 5\%) | --- | --- | 6.1E4 | 1.3E6 |
> > >
> > >
> > > **(2) Nonlinear extension**
> > > -   We agree that, in the nonlinear setting, omitting the Jacobian determinant ratio is a practical approximation and may introduce bias; we already note this limitation in the paper.
> > > -   Including the determinant ratio recovers exactness, but at additional cost.
> > >     This cost may be reduced using matrix-free randomized algorithms for computing the log-determinant  [1,2].
> > >     We will add this discussion in the revision.
> > >
> > > [1] Saibaba et al. “Randomized matrix-free trace and log-determinant estimators.” _Numerische Mathematik_, 2017.
> > > [2] Knoll and Keyes. “Jacobian-free Newton–Krylov methods: A survey of approaches and applications.” _JCP_, 2004.
> > >
> > >
> > > **(3) Scope of the theory**
> > >
> > > We agree that our theory is limited to the linear setting, as stated in the limitations; its role is to explain the mechanism of Proximal-IMH in a tractable regime. We will clarify this in the revision; extending the theory to nonlinear forward models is future work.
> > >
> > > **(4) Comparison with related methods**
> > > - Surrogate-based methods:   As noted in our rebuttal, the most directly related surrogate-based approaches are those that learn or approximate the forward operator. In our setting, such a surrogate naturally corresponds to the approximate operator $\widetilde{A}$. We therefore view these approaches as complementary rather than competing with our method: once a surrogate forward model is available, it can be used directly within Proximal-IMH as the approximate operator.
> > >
> > > - Learned proposal methods:
> > > 	 - To address this concern, we implemented a learned-proposal baseline using a **conditional normalizing flow** based on RealNVP [3] on a linear Gaussian inverse problem with the same exact operator $A$ as in Test I ($d_x=200$, $d_y=50$). The flow was trained amortized on $10^6$ simulated prior-observation pairs $(x,y)$, with standard normal prior and Gaussian noise.
> > > 	- For a fixed observation $y$, the learned posterior samples have about **15%** relative mean bias. We then used this learned posterior as an independence proposal in Metropolis--Hastings (**Flow-IMH**). For a fair comparison, we selected $\widetilde{A}$ so that the corresponding approximate posterior has a similar relative mean bias, about **13%**.
> > > 	- The _(# forward solves, relative mean error)_ results are shown in **Table 4**. In this test, **Flow-IMH** has a very low acceptance rate (**0.34%** over 30K IMH steps), behaves similarly to **Approx-IMH** since neither corrects the approximate proposal toward the exact posterior, and performs much worse than **Proximal-IMH**.
> > >
> > >
> > > **Table 4.** Cost–accuracy comparison across methods. Each entry reports _(# forward solves, relative mean error)_.
> > >
> > > | Approx-IMH | Latent-IMH | Proximal-IMH | Flow-IMH | NUTS |
> > > |:---:|:---:|:---:|:---:|:---:|
> > > | 1.0E5, 124\%| 9.0E5, 13.4\% | 1.0E4, 12.0\% | 2.0E5, 45.2\% | 2.9E5, 12.0\% |
> > > | 2.0E5, 92\% | 1.8E6, 10.9\% | 4.0E4, 6.8\% | 3.0E5, 35.4\% | 4.4E5, 7.5\% |
> > > | 3.0E5, 88\% | 2.7E6, 9.1\% | 1.9E5, 3.4\% | 5.0E5, 22.7\% | 1.4E6, 3.8\% |
> > >
> > >
> > > [3] Dinh, L., Sohl-Dickstein, J., and Bengio, S. Density estimation using Real NVP. ICLR, 2017.

---

### Decision · Program_Chairs · 2026-04-30

**Decision:**

Accept (regular)

**Comment:**

The paper introduces Proximal-IMH, which corrects proposals from an approximate posterior via local adjustments done by proximal optimisation in parameter space. Theoretical improvements (wrt Approx-IMH and Latent-IMH) are proven for linear Gaussian inverse problems. Experiments on bimodal linear targets, MNIST inverse problems, and Helmholtz PDEs show consistently higher acceptance rates and somewhat faster convergence than NUTS, with additional comparisons for flow-based IMH in the rebuttal.

The remaining concerns are that the theory only holds in Gaussian models. In the non-linear extensions, authors use the Gauss-Newton step while neglecting the Jacobian determinants, so exactness is no longer guaranteed. The authors admit that, yet still believe this is a practical and empirically validated approximation in their scenarios, and give a recipe on how to restore exactness at additional costs.

Overall, the method is conceptually clean, theoretically justified in some cases, and empirically strong. So I agree with the majority of the reviewers in their positive assessment and recommend weak accept for the paper.